**Investigation**

# Two coacting shadow enhancers regulate *twin of eyeless* expression during early *Drosophila* development

Jacqueline M. Dresch,[1] Luke L. Nourie [ID],[1] Regan D. Conrad,[1] Lindsay T. Carlson,[1] Elizabeth I. Tchantouridze,[1]
Biruck Tesfaye,[1] Eleanor Verhagen,[1] Mahima Gupta,[1] Diego Borges-Rivera,[1] Robert A. Drewell [ID] [1],*

[1]Biology Department, Clark University, 950 Main Street, Worcester, MA 01610, USA

*Corresponding author: Biology Department, Clark University, 950 Main Street, Worcester, MA 01610, USA. Email: rdrewell@clarku.edu

The *Drosophila* PAX6 homolog *twin of eyeless* (*toy*) sits at the pinnacle of the genetic pathway controlling eye development, the retinal determination network. Expression of *toy* in the embryo is first detectable at cellular blastoderm stage 5 in an anterior–dorsal band in the presumptive procephalic neuroectoderm, which gives rise to the primordia of the visual system and brain. Although several maternal and gap transcription factors that generate positional information in the embryo have been implicated in controlling *toy*, the regulation of *toy* expression in the early embryo is currently not well characterized. In this study, we adopt an integrated experimental approach utilizing bioinformatics, molecular genetic testing of putative enhancers in transgenic reporter gene assays and quantitative analysis of expression patterns in the early embryo, to identify 2 novel coacting enhancers at the *toy* gene. In addition, we apply mathematical modeling to dissect the regulatory landscape for *toy*. We demonstrate that relatively simple thermodynamic-based models, incorporating only 5 TF binding sites, can accurately predict gene expression from the 2 coacting enhancers and that the HUNCHBACK TF plays a critical regulatory role through a dual-modality function as an activator and repressor. Our analysis also reveals that the molecular architecture of the 2 enhancers is very different, indicating that the underlying regulatory logic they employ is distinct.

Keywords: *Drosophila*; *twin of eyeless*; enhancer; *cis*-regulation; development

## Introduction

The body plan in *Drosophila* is established in the early embryo by a network of transcription factors (TFs). Gradients of mRNAs and proteins maternally deposited in the egg sit at the very top of this genetic cascade that generates a positional coordinate system along the anterior–posterior (AP) and dorsal–ventral (DV) axes to divide the developing embryo into discrete domains (Ingham 1988). By embryonic stage 5 (cellular blastoderm), there are 6–23 cells allocated to form the eye-antennal primordium (Callaerts *et al.* 2006). *Pax* genes play critical roles in the development of the central nervous system (CNS), brain, and sensory organs, including the eye (Callaerts *et al.* 1997; Chi and Epstein 2002; Jacobsson *et al.* 2009). They encode for TFs characterized by a 128-amino acid paired DNA-binding domain. Several PAX proteins, including the evolutionarily conserved PAX6, have an additional DNA-binding homeodomain. In *Drosophila*, the two *Pax6* paralogs *twin of eyeless* (*toy*) and *eyeless* (*ey*) are at the pinnacle of the genetic pathway controlling eye development, the retinal determination (RD) gene network (Halder *et al.* 1995; Czerny *et al.* 1999; Jang *et al.* 2003). Previous studies have shown that expression of *toy* is first detectable at the cellular blastoderm stage in an anterior–dorsal band in the procephalic neuroectoderm (Czerny *et al.* 1999), which gives rise to the primordia of the visual system and brain (Kammermeier *et al.* 2001). In contrast, *ey* is first expressed during stage 10 (late germband extension) (Czerny *et al.* 1999). Biochemical and genetic data indicate that the TOY TF directly regulates *ey* by binding to an eye-specific enhancer located in

the second intron of the *ey* gene (Czerny *et al.* 1999; Hauck *et al.* 1999). The *cis*-regulatory control of *toy* expression in the early embryo is less well characterized.

Earlier studies by the Gehring lab were able to identify maternal and gap TFs involved in *toy* regulation in the early embryo (Blanco and Gehring 2008). Through extensive analysis of mutations in a number of genes, the maternally deposited and anteriorly localized *bicoid* (*bcd*) mRNA, which is translated to produce a gradient of BCD TF along the AP axis after fertilization, was found to act synergistically with the terminal system to activate *toy* in the anterior of the embryo (Blanco and Gehring 2008). In contrast, the HUNCHBACK (HB) and KNIRPS (KNI) gap TFs were identified as repressors, capable of restricting the temporal and spatial expression of *toy* along the AP axis (Blanco and Gehring 2008). Along the DV axis, peak levels of DECAPENTAPLEGIC (DPP) are necessary to restrict *toy* expression at the dorsal midline, and DORSAL (DL), largely acting indirectly through the positive regulation of KNI, is responsible for the repression of *toy* in the ventral half of the embryo (Blanco and Gehring 2008). The same group were able to identify a 300-bp embryonic eye primordium-specific enhancer (EEP) located 8.2 kb 5′ of the *toy* transcriptional start site, capable of driving reporter gene expression in stage 13 and 15 embryos in a pattern that co-localized with endogenous *toy* (Blanco *et al.* 2010). Genetic dissection of the promoter region also identified an additional 1.3-kb genomic fragment sufficient to drive *toy* expression in embryonic brain and eye primordia in stage 10 embryos (Jacobsson *et al.* 2009), although this result could not be reproduced by the Gehring group (Blanco *et al.* 2010). The existing

REDfly database of known and predicted enhancers in *Drosophila* (Rivera *et al.* 2019; Keränen *et al.* 2022) identifies up to 32 different *cis*-regulatory modules in the genomic regions surrounding the *toy* gene. With the exception of the EEP, almost all of these are not well defined (i.e. are over 3 kb in size) and/or drive expression in ubiquitous patterns not restricted to early development. Similarly, the Fly Enhancers database (Kvon *et al.* 2014) identifies six *cis*-regulatory modules at the *toy* locus, none of which drive specific patterns of expression in the early embryo.

A broad array of mathematical models have been used to study transcriptional regulation in the last two decades, with the objective of generating a better understanding of the molecular interactions occurring at predicted or known enhancers. One particular approach, the thermodynamic-based model, is commonly used to predict the transcriptional output of a target gene given the DNA sequence of the enhancer controlling expression and the concentrations for the TF proteins involved in regulation (Janssens *et al.* 2006; Dresch *et al.* 2013). To successfully do this, the TF binding sites within the enhancer are first identified and then all possible configurations of TF occupancy on the enhancer are enumerated. The gene expression output is predicted to be proportional to the ratio of the contribution of "successful" states of the enhancer to the contribution of all possible states of the enhancer (Spitz and Furlong 2012). Thermodynamic-based models can also account for the difference in activator and repressor activity and incorporate terms corresponding to important TF–TF interactions, such as cooperative binding or quenching of activator activity by short-range repressors (Dresch and Drewell 2012). For a complete derivation and review, see Dresch and Drewell (2012); Dresch *et al.* (2013).

Thermodynamic-based models have been used extensively in studies of gene regulation in *Drosophila* development (Janssens *et al.* 2006; Segal *et al.* 2008; Fakhouri *et al.* 2010; He *et al.* 2010; Parker *et al.* 2011; White *et al.* 2012; Kim, Martinez *et al.* 2013; Drewell *et al.* 2014; Sayal *et al.* 2016; Dibaeinia and Sinha 2021; Bhogale and Sinha 2022; Kim, Rhee *et al.* 2022). In 2008, Segal *et al.* utilized a thermodynamic-based model to predict gene expression from 44 different enhancer sequences and demonstrated that a simple model, in combination with a large amount of in vivo expression data, was capable of accurately predicting patterns of gene expression in the *Drosophila* embryo (Segal *et al.* 2008). In 2010, Fakhouri *et al.* implemented a more complex thermodynamic-based modeling approach on synthetic enhancer constructs to study mechanisms of short-range repression, or quenching, in the early *Drosophila* embryo (Fakhouri *et al.* 2010). In 2014, we used a similar model to investigate the regulatory architecture of the IAB7b enhancer in the bithorax cluster of homeotic genes and revealed the importance of modest protein–protein TF cooperativity in enhancer function (Drewell *et al.* 2014). More recently, Kim *et al.* used a model which included higher-order cooperative interactions and iteratively fit parameters using a synthetic enhancer library (Kim, Rhee *et al.* 2022). Higher-order cooperative interactions represent molecular interactions between more than two molecular players, including multiple regulatory proteins or proteins and RNA polymerase, and can account for a more complex grammar mediated by additional mechanisms such as the recruitment of co-factors, mediator complex, or other elements of the transcriptional machinery. Their study was the first in eukaryotes to show the necessity of accounting for higher-order interactions and showed that, in the case of an enhancer containing multiple binding sites for the RUNT repressor, the inclusion of protein–protein and higher-order cooperativity terms was essential to quantitatively predict

gene expression when 2 or 3 binding sites are present. One of the critical challenges to be addressed prior to implementing thermodynamic-based models lies in dissecting regulatory architectures to better understand the degree of complexity within a regulatory system, allowing such models to reach maximum predictive power.

For the regulation of the *toy* gene, the key unanswered questions are; how the complex *trans*-regulatory environment of the early embryo is able to establish the highly restricted anterior–dorsal band of *toy* expression, and where the genomic enhancers that control *toy* transcription are located. In this study, we take an integrated approach to address these two questions through the bioinformatic identification of novel enhancers or *cis*-regulatory modules (CRMs), testing of putative enhancers in transgenic reporter gene assays in embryos and quantitative analysis of expression patterns in the early embryo. In addition, we apply mathematical modeling to dissect the regulatory landscape for *toy*, specifically testing how complex a thermodynamic-based model needs to be to accurately predict gene expression from enhancers driving *toy* transcription and how much a simple model can teach us about the molecular interactions taking place at a single enhancer or coacting enhancers.

## Materials and methods
### Bioinformatic identification of putative *toy* enhancers

The RefSeq track (O'Leary *et al.* 2016) on the University of California Santa Cruz (UCSC) Genome Browser [http://genome.ucsc.edu (Kent *et al.* 2002)] was used to identify the location of the *twin of eyeless* (*toy*) gene within the *D. melanogaster* genome (BDGP Release 6 assembly) (Hoskins *et al.* 2015), and define the 31.5-kb region that spans from the *fussel* (*fuss*) gene upstream (5′) of *toy* to the *Plexin A* (*PlexA*) gene downstream (3′) of *toy*. Sequence conservation data was obtained from the MULTIZ (Blanchette *et al.* 2004) and phastCons (Siepel *et al.* 2005) tracks. Chromatin accessibility data were obtained from the *BDTNP Chromatin Accessibility (DNase) Replicate 1* track (Thomas *et al.* 2011). In vivo transcription factor binding data for BICOID (BCD), CAUDAL (CAD), DICHAETE (D), DORSAL (DL), HUNCHBACK (HB), KNIRPS (KNI), KRUPPEL (KR) and TWIST (TWI) were obtained from the *BDTNP ChIP–chip embryo stage* 4–5 track (Li *et al.* 2008; MacArthur *et al.* 2009).

### Computational analysis of transcription factor binding

Predicted TF binding sites in the putative *toy* enhancer zones were determined using Patser (Hertz and Stormo 1999) with previously assembled consensus Position Weight Matrices (PWMs) for BCD, CAD, D, DL, HB, KNI, KR and TWI using ln(*P*-value) cutoff values described in previous studies (Bergman *et al.* 2005; Meng *et al.* 2005; Noyes, Christensen, *et al.* 2008; Noyes, Meng, *et al.* 2008; Ho *et al.* 2009) and confirmed using the MAST algorithm (Bailey and Gribskov 1998; Bailey, Bodén *et al.* 2009).

### Transgenic reporter constructs

PCR primers were designed to amplify putative *toy* enhancer zones (see Table 1). PCR amplicons were cloned into pGEM®-T Easy vector (Promega), then sub-cloned as *Not*I fragments into the *placZattB* transformation vector (Bischof *et al.* 2007) and subsequently sequenced to check for insertion orientation. Constructs containing putative *toy* enhancer zones were introduced into the *D. melanogaster* germline via φC31-mediated integration (Bischof

**Table 1.** PCR Primers designed to amplify putative *toy* enhancer zones .

| Zone | Primer | Primer sequence (5′->3′) | Genomic location (chr 4) | Product size (bp) |
| --- | --- | --- | --- | --- |
| 1 | S | ATTGAAGTTACATCTTATAGTTTGA | 978362 | 2319 |
|  | AS | ACTCAGCTAATATGTGCGAA | 980680 |  |
| 2 | S | ATCTTGGTGTATGTGAAAACGGA | 980855 | 1174 |
|  | AS | TCCACTAACTGAGTAACGGGTAA | 982028 |  |
| 4 | S | TGATAGTGGGGTTCGGTTAGT | 984617 | 1169 |
|  | AS | AAGGAAAATGCATATATTAAGCTGT | 985785 |  |
| 7 | S | GATGGGGAATTGATACAGGGCA | 989115 | 806 |
|  | AS | TTTCCGTCGAAAGTACGGGTC | 989920 |  |

*et al.* 2007). All microinjections were carried out by BestGene Inc., using the *attB* 68E site for targeted integration of the reporter constructs as previously described (Drewell *et al.* 2014).

## *In situ* hybridization and imaging

Embryos from all eight transgenic *D. melanogaster* lines were collected, fixed, and hybridized with a digoxigenin-labeled *lacZ* probe as previously described (Drewell *et al.* 2014). A minimum of 8 embryos from each line ($8 < n < 17$) were imaged on a Zeiss Axio Imager 2 microscope with a 0.9 HD Ph DIC condenser module by a Canon EOS Rebel T2i camera and native Zeiss software at 20× objective magnification, using z-stack imaging with 5.2V intensity and 23-millisecond exposure time. For each embryo, positioned in a lateral view, 20–35 non-overlapping 1.5-μm optical sections were captured, representing the proximal half of each embryo. The developmental stage and orientation of each embryo were determined using characteristics such as the appearance of pole cells at the posterior of the embryo, symmetry of the embryo, and the presence or absence of marks of gastrulation (Weigmann *et al.* 2003).

## Image processing pipeline for zones 1 and 2 in stage 5 embryos

All images were processed in a 6-step procedure involving binary image generation, rotation, resizing, background subtraction, normalization, and intensity-value extraction. Binary image generation, rotation, and resizing were done as described previously (Sayal *et al.* 2016). For background subtraction, we used a uniform background subtraction method by averaging the intensity values in a non-expressing region of the embryo and subtracting that value from all points within the boundaries of the embryo. For this study, the non-expression region used was defined as the region from 45% to 55% on the AP axis and from 20% to 50% on the DV axis. Normalization was performed as described previously (Sayal *et al.* 2016), using zone 2F embryos for normalization.

The area of interest for all embryos comprised a region at 25% DV, spanning from 10 to 95% AP (see Supplementary Fig. 1). We fixed the location on the DV axis to enable our modeling efforts to focus on the regulatory roles of the previously identified maternal and gap TFs across the AP pattern of expression driven by the putative *toy* enhancers. The 25% DV position was selected based on the observation that endogenous *toy* and putative enhancer-driven *lacZ* expression exhibit peak expression between 21 and 28% DV.

To get a single line of intensity values from this region, while minimizing the noise from sampling a single line of pixels, we calculated an "average nucleus intensity" value for each pixel. It was previously determined that the length of the AP axis from a lateral view of a developing stage 5 *D. melanogaster* embryo is approximately 65 nuclei (Fowlkes *et al.* 2008; Drewell *et al.* 2014). Thus, the diameter of a nucleus in pixels was calculated using the following formula:

$$\text{Nucleus diameter} = \frac{\text{Number of pixels across AP axis}}{65}$$

The average pixel intensity was then calculated using a one-nucleus neighborhood:

$$\text{Average pixel intensity}$$
$$= \frac{\text{Sum of pixel intensities within a 1 nucleus radius}}{\text{Number of pixels within the neighborhood}}$$

For each transgenic line analyzed quantitatively, the average intensity profiles and standard error calculations across all embryos imaged from that transgenic line are shown in Fig. 6.

## Thermodynamic-based modeling

All modeling was done using a thermodynamic-based model, as described in our previous studies (Dresch and Drewell 2012). The transcriptional output from the zone 1 and zone 2 enhancers were modeled separately. For each enhancer, binding sites were predicted using the MAST algorithm (Bailey and Gribskov 1998; Bailey, Bodén *et al.* 2009) for five transcription factors shown to regulate the *toy* gene across the AP axis; BCD, CAD, HB, KNI, and KR (Blanco and Gehring 2008). In an effort to find the simplest model that produced a fit to the experimental expression data, we included up to two binding sites per TF, to allow for the incorporation of TF protein–protein interaction parameters, and explored the complete model space.

All models included a scaling parameter (*K*-value) for the efficiency/strength of each TF binding site and parameters for each possible pair of interactions between TF binding sites. Within the model, *K*-values are binding site specific and are multiplied by each term representing a state of the enhancer in which that specific binding site is occupied. The interpretation, at a molecular level, is that this term represents the efficiency or strength of that TFBS, as the larger the *K*-value the more weight a state with that specific site bound is given. Interaction terms were incorporated into the model in states where the pair of TFs were simultaneously and adjacently bound (i.e. both TFs are bound with no other TFs bound in between the 2 sites on the enhancer). Since cooperativity (*C*) values are multiplied on to each state in which the 2 TFBSs are bound simultaneously, one can interpret the interaction as positive cooperativity/synergistic activity when $C > 1$, anti-cooperative/competitive activity when $C < 1$ and non-cooperative/independent activity when $C = 1$. Since quenching parameters, representing short-range repression, can be interpreted as the proportion of the state that contributes to activation when both TFs are bound, all quenching (*Q*) values are between 0 and 1. TF scaling factors (*K*) and cooperativity (*C*) values were allowed to take on any number greater than or equal to 0.

As it is unclear what the role of HB is in regulating *toy*, when modeling enhancers containing at least one HB binding site, we

included additional parameters to allow for potential dual-regulatory (activator and repressor) activity. These parameters included one threshold parameter, which represents a position on the AP axis at which HB regulation switches from activation to repression (or vice-versa), and additional scaling and interaction parameters for each HB site to be used in positions along the AP axis in which HB activity is switched. The HB scaling parameters are denoted in our results as *K*-values corresponding to a specific HB site (i.e. in Fig. 8, HB1(A) and HB1(R)).

TF concentrations were taken from the BDTNP dataset, which contains three-dimensional (3D) measurements of relative protein and mRNA concentration for a large number of genes. For the TFs used in this study, we were able to obtain protein concentrations for BCD, HB, and KR at 80 min into development and mRNA concentrations for *cad* and *kni* at 70 min into development (Fowlkes *et al.* 2008). For CAD and KNI, we used the mRNA concentrations at 70 min into development to predict protein concentrations at 80 min into development using a simple translation model (Jaeger *et al.* 2004).

## Conservation of enhancer sequences across *Drosophila* species

Bioinformatic analysis of the zone 1 and zone 2 genomic regions was performed as previously described (Drewell *et al.* 2014). For each enhancer, putative TF binding sites were determined using the MAST algorithm (Bailey and Gribskov 1998; Bailey, Bodén *et al.* 2009) with the previously assembled PWMs for BCD, CAD, HB, KNI, and KR (Bergman *et al.* 2005; Meng *et al.* 2005; Noyes, Christensen, *et al.* 2008; Noyes, Meng, *et al.* 2008; Ho *et al.* 2009) using ln(*P*-value) cutoff values equal to those of the weakest of the two binding sites for each TF (used in the models described in the previous section) within each zone.

## Results

### Bioinformatic analysis to identify putative toy enhancers

To identify putative embryonic enhancers for the *twin of eyeless* (*toy*) gene, we investigated the 31.5-kb genomic region surrounding *toy*. This region extends from the *fussel* (*fuss*) gene upstream (5′) of *toy* to the *Plexin A* (*PlexA*) gene downstream (3′) of *toy* (Fig. 1). We examined the entire region for evolutionary conservation, chromatin accessibility and in vivo TF binding (see Methods for details) and

identified 16 genomic zones that displayed evidence of at least 2 of these features (Fig. 2). Zones 1–6 are located upstream of the *toy* gene, zone 7 encompasses the promoter region and transcriptional start site for the two *toy* transcript isoforms, zones 8–13 are located in introns of the *toy* gene, zones 14 and 15 are predominantly in exons located at the 3′ end of the gene, and zone 16 is located downstream of the *toy* gene (Fig. 1). Zone 2 encompasses the previously characterized 300-bp embryonic eye primordium-specific (EEP) enhancer (Blanco *et al.* 2010) located 8.2-kb upstream of the *toy* transcriptional start site (Fig. 1).

As the function of an enhancer is conferred by the binding of specific TFs to the genomic DNA sequence, measurable in vivo binding by 8 of the TFs [BICOID (BCD), CAUDAL (CAD), DICHAETE (D), DORSAL (DL), HUNCHBACK (HB), KNIRPS (KNI), KRUPPEL (KR), and TWIST (TWI)] known to play a role in regulating *toy* expression (Blanco and Gehring 2008) was considered a potentially important feature of the 16 putative enhancer zones. Accordingly, only 3 of the zones (5, 12, and 13) fail to show any evidence of binding amongst the 8 TFs we examined in stage 4–5 embryos (Fig. 2b). In addition, we took into account chromatin accessibility at the putative enhancers, as TFs require open chromatin environments for binding events to occur. All 16 zones have some degree of chromatin accessibility at stage 5 of development, with many persisting into later developmental stages (Fig. 2c). To further investigate the putative *toy* embryonic enhancers, we searched all 16 zones for predicted binding sites for the 8 TFs described above using Patser, a common PWM-based search algorithm (Hertz and Stormo 1999). The results are shown in Supplementary Fig. 2 and summarized in Supplementary Table 1. Patser predicts binding sites for the majority of TFs in all 16 zones, with the total number of sites ranging from 12 to 99 (Supplementary Table 1).

Integrating all our bioinformatic analysis allows us to assess all four criteria (sequence conservation, in vivo TF binding, chromatin accessibility, and predicted TF binding sites) in parallel for the 16 putative *toy* embryonic enhancer zones (Fig. 3). The summary data reveal that zones 1, 2, 4, and 7 contain sub-regions that demonstrate relatively strong sequence conservation, as measured by phastCons, when compared to surrounding genomic regions, and are evolutionarily conserved across many *Drosophilid* species in MULTIZ alignments. All four of these zones also show moderate to high chromatin accessibility in embryos, with a notably high signal detected in zones 1 and 7. ChIP/chip data indicate in vivo binding at zones 1, 2, 4, and 7 for at least two of the 8 TFs examined in stage 4–5 embryos. In addition, analysis of predicted binding

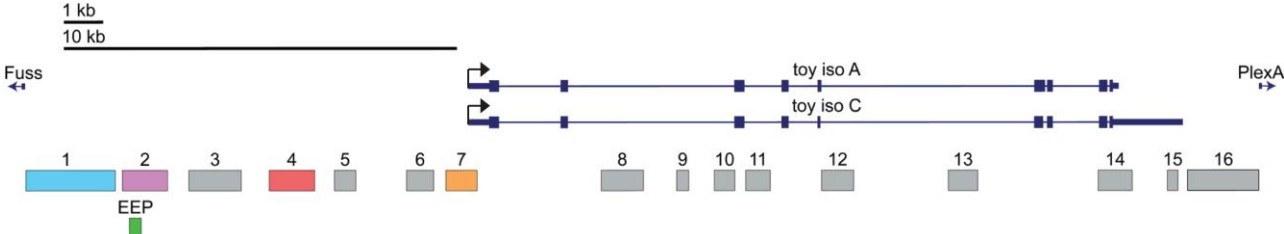

**Fig. 1.** *Toy* putative enhancers. Map showing the locations of the *toy* gene and the putative enhancer zones identified through bioinformatic analysis. The 31.5-kb genomic region depicted is located on chromosome 4, sequence coordinates 978,397–1,009,894 (*D. melanogaster* genome release6). The *fussel* (*fuss*) gene upstream of *twin-of-eyeless* (*toy*) and the *Plexin A* (*PlexA*) gene downstream of *toy* are indicated. The two annotated isoforms of *toy*, toy iso A and toy iso C, are show with their exon (thick dark blue box), intron (thin dark blue line), and 5′ UTR and 3′UTR (medium dark blue box) structure. Arrows indicate the transcriptional start site for *toy*, which is identical for the two isoforms. The 16 zones identified through bioinformatic analysis are numbered below the genes. The previously identified EEP (embryonic eye primordium-specific) enhancer is shown in green. The four zones selected for further study in vivo are colored (1—light blue, 2—purple, 4—red, 7—orange).

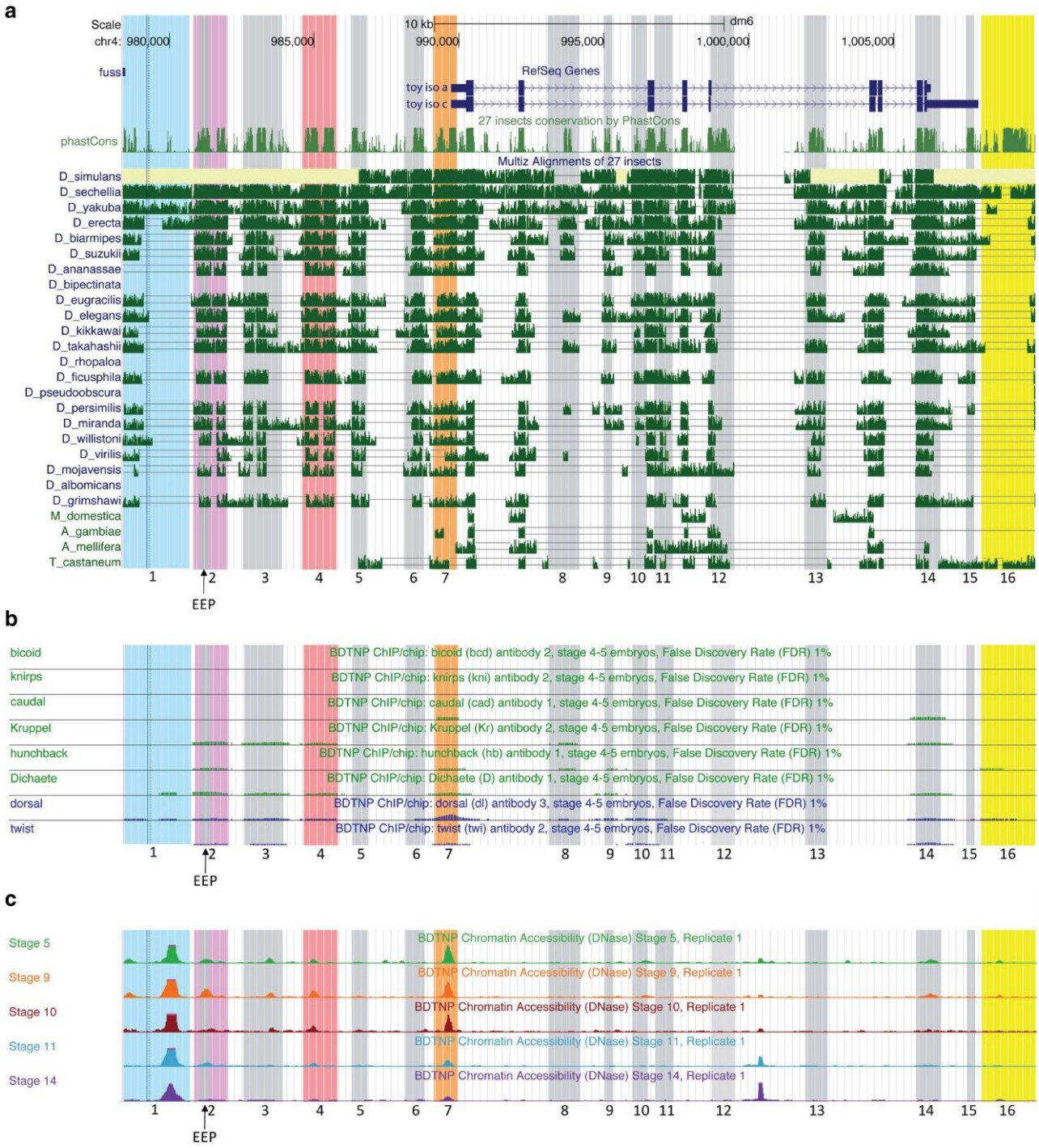

**Fig. 2.** *Toy genomic region features.* a) Sequence conservation in phastCons and MULTIZ tracks. The 31.5-kb genomic region around the *twin-of-eyeless* (*toy*) gene analyzed in the UCSC Genome browser is located on chromosome 4, sequence coordinates 978,397–1,009,894 (*D. melanogaster* genome release6). The two annotated isoforms of *toy*, toy iso A and toy iso C, are shown with their exon (thick dark blue box), intron (thin dark blue line), and 5′ UTR and 3′ UTR (medium dark blue box) structure. Arrows indicate the direction of transcription for the *toy* gene. In the phastCons and MULTIZ alignment tracks, the height of the bar indicates likelihood that a particular genomic sequence is conserved. b) In vivo transcription factor ChIP/chip binding data. Transcription factor (TF) binding in stage 4–5 embryos from the Berkeley Drosophila Transcription Network Project (BDNTP) for 6 TFs implicated in AP patterning (green) and 2 TFs implicated in dorsal–ventral patterning (blue). The height of the bars in each track indicates the relative strength of TF binding in genomic sequence. c) Chromatin accessibility data. Chromatin accessibility, measured by *DNase I* digestion, from the BDNTP at 5 different embryonic stages of development. The height of the bars in each track indicates relative accessibility of the genomic region. The 16 zones identified through bioinformatic analysis are numbered below the genes. The previously identified EEP (embryonic eye primordium-specific) enhancer, located within zone 2, is also indicated with an arrow. The four zones selected for further study in vivo are colored (1—light blue, 2—purple, 4—red, 7—orange).

| Zone | Size (bp) | phastCons | MULTIZ | Chromatin accessibility | TF binding | Patser |
|------|-----------|-----------|--------|-------------------------|------------|--------|
| Zone 1 | 2304 | moderate | moderate | high | 2 | 8 |
| Zone 2 | 1243 | high | high | moderate | 5 | 8 |
| Zone 3 | 1341 | high | high | moderate | 4 | 7 |
| Zone 4 | 1087 | high | high | moderate | 3 | 7 |
| Zone 5 | 551 | high | high | moderate | 0 | 7 |
| Zone 6 | 688 | high | high | low | 1 | 8 |
| Zone 7 | 806 | high | high | high | 5 | 7 |
| Zone 8 | 1094 | moderate | low | none | 3 | 8 |
| Zone 9 | 309 | high | low | none | 3 | 5 |
| Zone 10 | 529 | moderate | low | moderate | 2 | 7 |
| Zone 11 | 634 | moderate | high | none | 2 | 8 |
| Zone 12 | 817 | high | high | none | 0 | 8 |
| Zone 13 | 764 | high | high | low | 0 | 8 |
| Zone 14 | 883 | high | high | moderate | 5 | 7 |
| Zone 15 | 280 | high | moderate | none | 1 | 5 |
| Zone 16 | 1861 | high | low | low | 2 | 8 |

**Fig. 3.** Summary of bioinformatic analysis. phastCons—likelihood that conserved elements are contained in each zone. MULTIZ—how well each zone is conserved through evolutionary time. Chromatin accessibility—level of chromatin accessibility, inferred from susceptibility to DNaseI digestion. ChIP/chip—number of transcription factors of interest, out of a total of eight, for which a ChIP/chip assay identified in vivo binding within a zone. Patser—number of transcription factors of interest, out of a total of eight, for which Patser predicted binding sites. Shading—green, yellow, blue, and gray shading correspond to high, medium, low, or no level of the particular criteria measurement, respectively. For the ChIP/chip and Patser numeric criteria, sites for 1 to 3 factors were considered low, 4 to 6 were considered moderate, and 7 or 8 was considered high. The 4 zones selected for further analysis are highlighted in red.

sites for the TFs reveals that all 4 zones have potential binding sites for 7 or 8 of these TFs. Amongst the other 12 zones, no single zone met all the shared criteria observed for zones 1, 2, 4, and 7 (Fig. 3). Taken as a whole, the bioinformatic data suggest that zones 1, 2, 4, and 7, all of which lie upstream (5′) of the *toy* gene, are the most likely to harbor embryonic enhancers. We therefore selected these four genomic zones for further analysis.

## Qualitative analysis of embryonic expression from putative enhancers

To test the potential functional activity of zones 1, 2, 4, and 7, the corresponding genomic region for each zone was individually cloned into a transgenic *lacZ* reporter construct in both forward (F) and reverse (R) orientations. The constructs were then integrated in a site-specific manner into the *D. melanogaster* genome and reporter gene expression driven by each zone was visualized by in situ hybridization in transgenic embryos (Fig. 4). At stage 5, no expression was detected in embryos from lines 4F, 4R, 7F, and 7R (Fig. 4, e–h), suggesting that neither zone 4 nor 7 contain regulatory sequences capable of acting as embryonic enhancers for the *toy* gene. In contrast, in stage 5 embryos from lines 1F, 1R, 2F, and 2R expression was observed (Fig. 4, a–d). Similar expression patterns were observed between embryos, irrespective of the zone or the orientation of the zone on the transgene, indicating that both zones 1 and 2 can drive embryonic gene expression in an orientation-independent manner and therefore fulfill the classic definition of an enhancer (Banerji *et al.* 1981).

The similar expression pattern observed in embryos from lines 1F, 1R, 2F, and 2R recapitulates the previously described expression pattern of endogenous *toy* (Czerny *et al.* 1999). The spatio-temporal pattern of both zone 1 and zone 2-driven *lacZ* reporter

gene expression was analyzed during a time course of *Drosophila* development and found to be qualitatively indistinguishable from each other. Comparison to the reported pattern (Czerny *et al.* 1999) of *toy* expression reveals very similar overlapping patterns of expression (Fig. 5). During embryogenesis, transcripts of both the *toy* gene and *lacZ* reporter are first detected at the cellular blastoderm stage in the posterior procephalic region including the presumptive optic lobe area (Fig. 5, a–d). As development progresses, the *toy* expression domain in this dorsolateral head ectoderm region gives rise to the brain and most of the visual system (Green *et al.* 1993; Younossi-Hartenstein *et al.* 1993) including the optic lobe, the larval eyes (Bolwig's organ), and eventually the eye imaginal discs from which the adult compound eyes and the 3 ocelli develop. In gastrulating embryos, *toy* and *lacZ* expression is confined to the head region anterior to the cephalic furrow (Fig. 5, e and f). Following germband retraction, expression is detected in the optic lobe primordia (Fig. 5, g–j, arrowheads) and in a broad region of the developing brain. At this stage, *toy* and *lacZ* expression is also observed in a segmentally reiterated pattern in the ventral nerve cord (Fig. 5, k and l).

In cellular blastoderm (stage 5) embryos, reporter gene expression is detectable in a spatially restricted anterior–dorsal region (indicated with red arrow in Fig. 5d). This overlaps with the area of the embryo where endogenous *toy* is expressed at the same developmental stage (Fig. 5c) (Czerny *et al.* 1999), and suggests that zones 1 and 2 are potentially involved in the regulation of early embryonic expression of *toy*. However, the expression patterns from zones 1 and 2 also appear to extend to the ventral side of the embryo and there is an additional posterior band of expression visible in many embryos (Fig. 4, a–d). Furthermore, visual qualitative analysis of stage 5 embryos indicates that the level of *lacZ*

Forward | Reverse

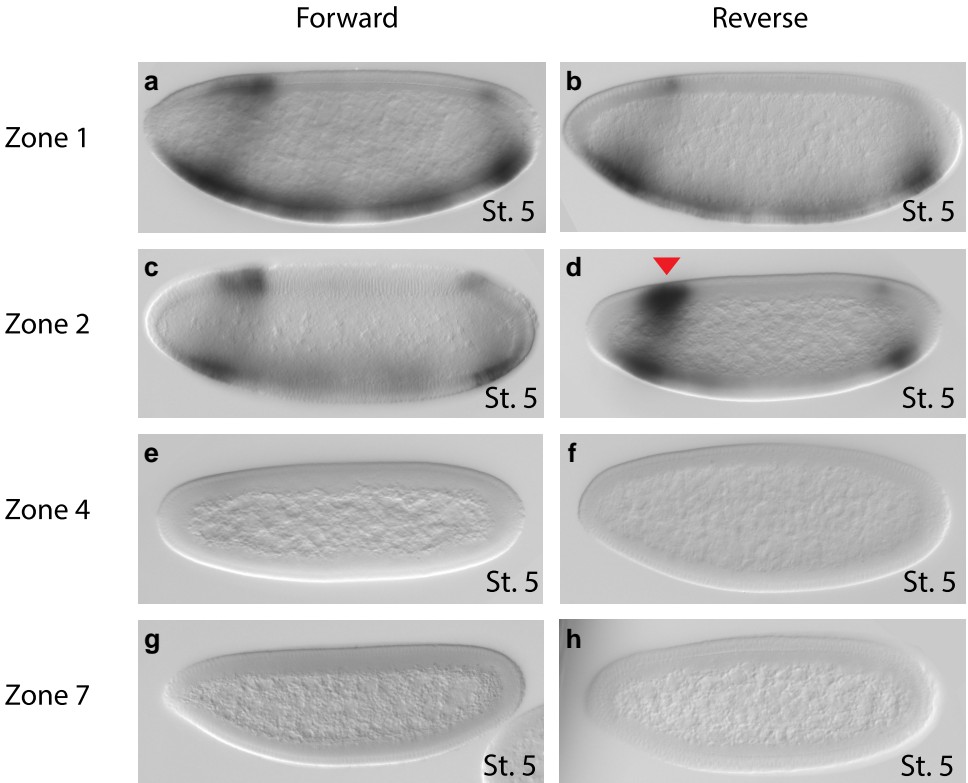

**Fig. 4.** Embryonic stage 5 expression patterns. Stage 5 (cellular blastoderm) transgenic embryos from zones 1, 2, 4, and 7 assayed with an anti-sense *lacZ* probe. Embryos with the putative enhancer in the forward (a, c, e, and g) or reverse (b, d, f, and h) orientation are shown. Red arrowhead indicates the anterior–dorsal area of expression in the same location as previously reported endogenous *toy* expression. Additional domains of *lacZ* expression are observed in the anterior–ventral, posterior–dorsal and posterior–ventral regions of the embryo in zone 1 and 2 embryos (a–d), while no *lacZ* expression is detected in zone 4 and 7 embryos (e–h).

expression is higher from zone 2, when compared to zone 1. In contrast, no reporter gene expression was detectable in embryos carrying zone 4 or zone 7 at any stage of embryonic development. The discovery of two potential embryonic enhancers for *toy* and their similar, but not identical, complex expression patterns led us to turn to a more quantitative analysis of the expression from zones 1 and 2 at stage 5 of embryonic development.

## Quantitative analysis of expression from zones 1 and 2

To investigate the differences in reporter gene expression driven by zones 1 and 2 we created an image processing pipeline to analyze reporter gene expression in stage 5 embryos after uniform in situ hybridization staining with an anti-sense *lacZ* probe. The pipeline included steps for image rotation and resizing, background subtraction, and normalization (see Methods for full details). After images were processed, we implemented a detailed quantitative analysis of the expression patterns within the embryo along the AP and DV axes. All statistical analyses and mathematical modeling of transgenic embryos containing zone 1 or zone 2 were performed using expression across the AP axis at 25% DV, as this is where the highest expression was found in the embryo (see Methods and Supplementary Fig. 1 for further details). Pearson correlation coefficients, calculated for pairwise comparisons between *lacZ* reporter gene transcripts from lines 1F, 1R, 2F, and 2R, show that the expression patterns in stage 5 embryos are highly correlated (>0.81) for all zones and orientations (Fig. 4 and Table 2). The highest correlation values are found when comparing the two orientations of the same zone (0.92 for

zone 1 and 0.96 for zone 2), further indicating that both zones fulfill the orientation-independent activity requirement of a defined enhancer (Banerji *et al.* 1981).

## Enhancers drive statistically significant different patterns of anterior expression

While both zone 1 and 2 embryos exhibit a broadly similar expression pattern characterized by a clear anterior peak in addition to a weaker posterior peak (Fig. 4), there are quantitative differences. At 25% DV, the location of the anterior peak of expression varies from embryo to embryo, ranging from 17 to 34% AP, with an overall markedly sharper peak of expression from zone 2, compared to zone 1 (Fig. 6). The average peak of expression for zone 1 is at 26.04% AP and for zone 2 is at 21.30% AP. This variation in the position of peak expression from zone 1 to zone 2 is not statistically significant (Mann–Whitney *U*-test with $\alpha = 0.05$). Comparison of the level of expression at 25% DV between the two enhancers reveals that the peak expression in zone 1 embryos is significantly lower than the peak expression level in zone 2 embryos (Fig. 6, Mann–Whitney *U*-test, all pairwise *P*-values < 0.005). The average normalized peak expression from zone 1F is 0.2801 and from zone 1R is 0.3498, compared to values of 0.7936 from zone 2F and 0.6831 from zone 2R. Notably, at 25% DV, neither the zone 1F and 1R pair, nor the zone 2F and 2R pair, differ significantly from each other at an $\alpha = 0.05$ significance level. The statistical significance of the difference in the expression levels between the zones, but not between the orientations of a single zone, suggests that each zone is acting as an independent enhancer capable of driving nonidentical spatial patterns of anterior expression in cellular blastoderm stage 5 embryos.

toy                                     Zone 2-lacZ

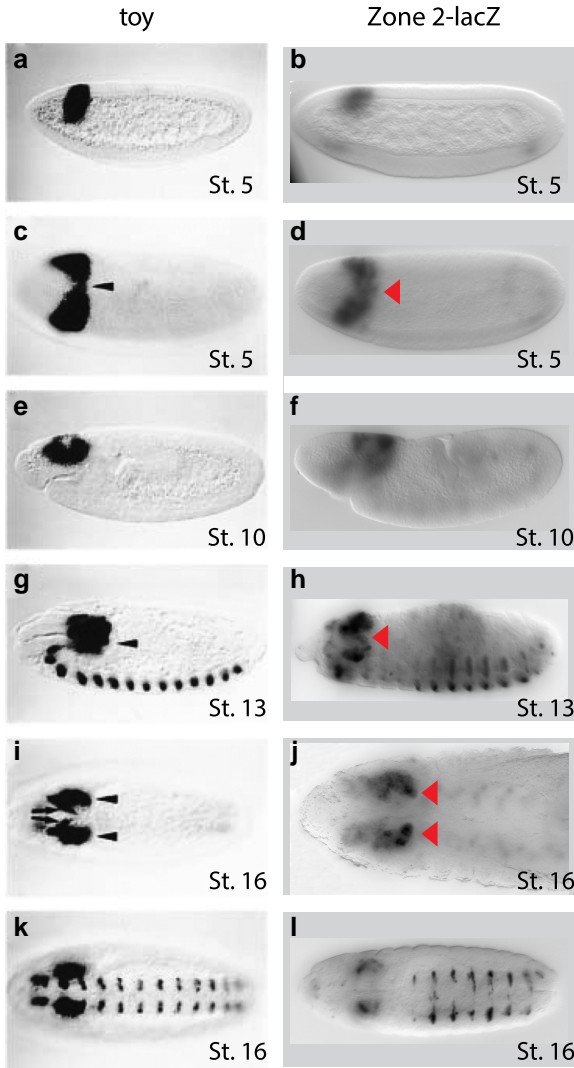

**Fig. 5.** Developmental expression profile for toy gene and zone 2 enhancer-*lacZ*. Wholemount in situ hybridization of developmental stage-matched embryos showing the spatiotemporal pattern of endogenous toy (left) or zone 2 enhancer-driven *lacZ* (right) expression. (a and b) Lateral view of cellular blastoderm (stage 5) embryo. (c and d) Dorsal view of stage 5 embryo, arrowheads point to dorsal midline. (e and f) Lateral view of germband extension (stage 10) embryo. (g and h) Lateral view of germband retraction (stage 13) embryo, arrowheads point to optic lobe. (i and j) Dorsal view of embryo after head involution (stage 16), arrowheads point to optic lobe. (k and l) Ventral view of stage 16 embryo. All toy expression data is from Czerny et al. (1999).

## Modeling enhancer activity

In an effort to further dissect the regulatory logic of the two toy enhancers, we employed mathematical modeling approaches. Specifically, we implemented proven thermodynamic-based models (Janssens et al. 2006; Segal et al. 2008; Fakhouri et al. 2010; He et al. 2010; Parker et al. 2011; White et al. 2012; Kim, Martinez et al. 2013; Drewell et al. 2014; Sayal et al. 2016; Dibaeinia and Sinha 2021; Bhogale and Sinha 2022; Kim, Rhee et al. 2022) to investigate the functional role of predicted transcription factor binding sites (TFBSs), a range of parameters relating to the molecular activity of the regulatory TFs, and the overall complexity of the enhancers. Our overarching goal was to identify the key molecular components regulating the transcriptional output from the two enhancers. To achieve this, we initially focused on

**Table 2.** Pearson correlation coefficients for all pairwise comparisons of zones 1F, 1R, 2F and 2R at 25% dorsal–ventral axis

|     | 1F | 1R     | 2F     | 2R     |
| --- | --- | --- | --- | --- |
| **1F** | 1 | 0.9196 | 0.9084 | 0.8412 |
| **1R** | x | 1      | 0.9039 | 0.8174 |
| **2F** | x | X      | 1      | 0.9554 |
| **2R** | x | X      | x      | 1      |

The matrix is symmetrical over the diagonal, so identical values have been replaced with an x.

creating the simplest possible thermodynamic-based models capable of accurately predicting the quantitative expression profiles from zones 1 and 2 at 25% DV across the AP axis from 10–90% AP in stage 5 embryos (see Methods for full details). Given the characterized orientation-independent activity of each zone, we averaged the expression profiles for zones 1F and 1R, as well as expression profiles for zones 2F and 2R, to generate a single unified profile for the zone 1 enhancer and a single unified profile for the zone 2 enhancer for use in model fitting.

Bioinformatic analysis was utilized to identify the two strongest predicted binding sites in each enhancer for each of the five AP patterning TFs previously shown to play a role in regulating toy expression; BICOID (BCD), CAUDAL (CAD), HUNCHBACK (HB), KNIRPS (KNI) and KRUPPEL (KR) (Blanco and Gehring 2008). The resulting 10 predicted binding sites in each of the two enhancers (Fig. 7, a and b) enable the analysis of relatively simple models for each enhancer zone, while allowing for all possible TF–TF interactions. The inclusion of the 2 strongest binding sites per TF was implemented so we could explore all possible repressor–activator combinations as well as the possibility of both heterotypic and homotypic protein–protein cooperativity impacting the transcriptional output of each enhancer. To exhaustively explore model space independently for each of the two zones, we fit 1,024 models for each enhancer, including every possible subset of TFBSs within each zone (see Methods for full details on the model implementation). Each of these models represents an enhancer containing a subset of 10 TFBSs, six of which are strictly repressor binding sites for CAD, KNI and KR.

## Increasing the number of binding sites does not imply better model performance

Although adding more binding sites to a thermodynamic-based model generally leads to a lower error, the results from the 2,048 model variations in our study show that the relationship between the number of TFBSs included in a model and the model error, as measured by root mean square error (RMSE), is not strictly negative (Fig. 7, c and d). However, due to the variation in complexity amongst the models, the number of parameters also varies greatly from model to model. To account for this, we also calculated Akaike Information Criterion (AIC) values for model comparison. The AIC is a standard measure of error for model comparison which includes a penalty term for the number of parameters; thus, providing a single measure of the tradeoff between goodness of fit and model simplicity. Strikingly, for both zones 1 and 2, we find an overall similar pattern to the RMSE analysis, namely that an increase in the number of TFBSs does not necessarily increase the performance of the model (Fig. 7, e and f), and discover that the models that result in the absolute minimum AIC values contain only five of the 10 predicted TFBSs (min AIC values: zone 1 = −389.92, zone 2 = −245.58). In the case of the zone 1 enhancer the five TFBSs represent one binding site for each of the

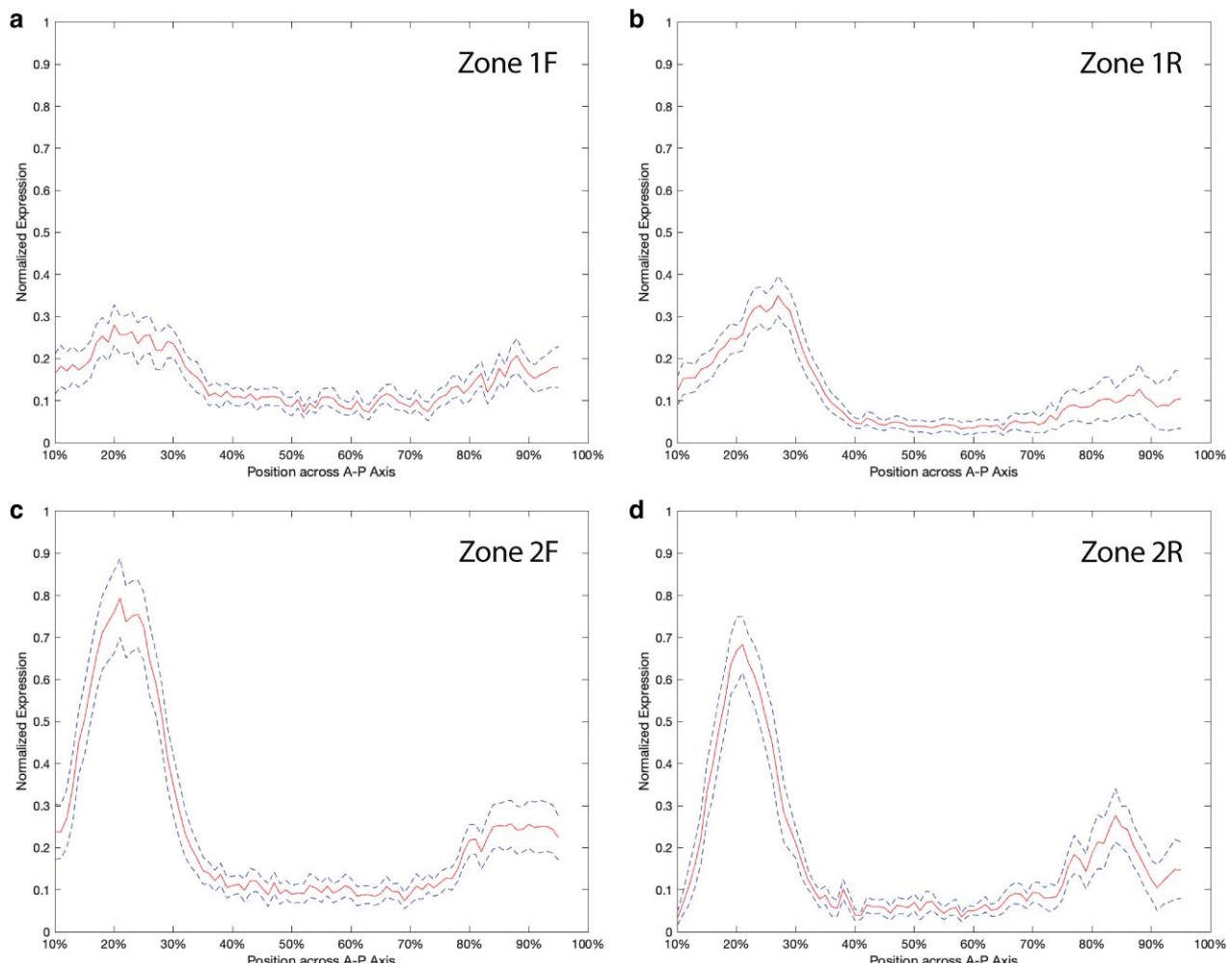

**Fig. 6.** Quantitative analysis of enhancer-driven expression in stage 5 embryos. Intensity plots from 10 to 95% anterior–posterior at 25% dorsal–ventral for zones 1F (a, $n = 17$), 1R (b, $n = 11$), 2F (c, $n = 12$) and 2R (d, $n = 8$). Solid red lines are mean expression. Dashed blue lines indicate standard error.

five different TFs (Fig. 7, g and i), while for the zone 2 enhancer, there are two BCD binding sites, and one each for HB, KNI and KR (Fig. 7, h and j).

## Contribution of individual binding sites

Discovering that the best-performing model for each zone only included five total TFBSs on the enhancer, we turned our attention to uncovering the relative importance of each individual binding site in regulating the transcriptional output from each enhancer. Examining the $K$-values (the parameters representing the efficiency/strength of an individual TFBS) for the 10 binding sites considered in the models reveals that the median value is below 15, and in most cases is below 10, for all TFBS in both enhancers, but there is a wide range of values for every site (Fig. 8, a and b). The two BCD activator binding sites have the highest median $K$-values in the zone 1 enhancer (Fig. 8a). Likewise, the 2 BCD activator, and also the 2 HB activator, binding sites have the highest median values in the zone 2 enhancer (Fig. 8b). Perhaps unsurprisingly, this indicates that the activator TFBSs are relatively important for robust model performance.

If we examine the $K$-values within each of the 1,024 different models in hierarchically clustered heatmaps, some interesting patterns emerge. Notably, in models where a high $K$-value is reported for any one of the four activator binding sites (BCD1, BCD2, HB1(A), HB2(A)), there is not a corresponding high $K$-value

for any of the other 3 activator binding sites. This pattern is consistent for both the zone 1 enhancer (Fig. 8c) and the zone 2 enhancer (Fig. 8d) and suggests that expression from each enhancer is driven by a dominant activator binding site with other binding sites working to refine expression through cooperative interactions. In addition, the relatively low $K$-values consistently observed across the majority of models for all the CAD, KNI, and KR repressor TFBSs indicate that there is not a dominant binding site (or sites) amongst them. Rather, the global pattern of $K$-values across the models suggests a potentially complex landscape for repression at both the zone 1 and zone 2 enhancers, with repressors working in concert to modulate the spatial output in the embryo. These observations are supported when we look at the details of the best-performing models for each zone.

For the zone 1 enhancer, the best-performing model (measured by AIC) resulted in an RMSE of approximately 0.01 when compared to the experimental expression data (Fig. 7, g and i). The parameter values resulting in this RMSE contained relatively large $K$-values ($>10$) for both the BCD2 and HB2 activator sites, but low $K$-values ($1 < K < 3.5$) for the 3 (CAD2, KNI1 and KR2) repressor sites (Supplementary Table 2). However, the cooperativity ($C$) values representing the pairwise interactions for all of the repressor sites were much larger than those found for the activator sites. Quenching ($Q$) values were relatively small ($<0.3$) for all repressor combinations with the BCD2 site and the quenching of KNI1 on

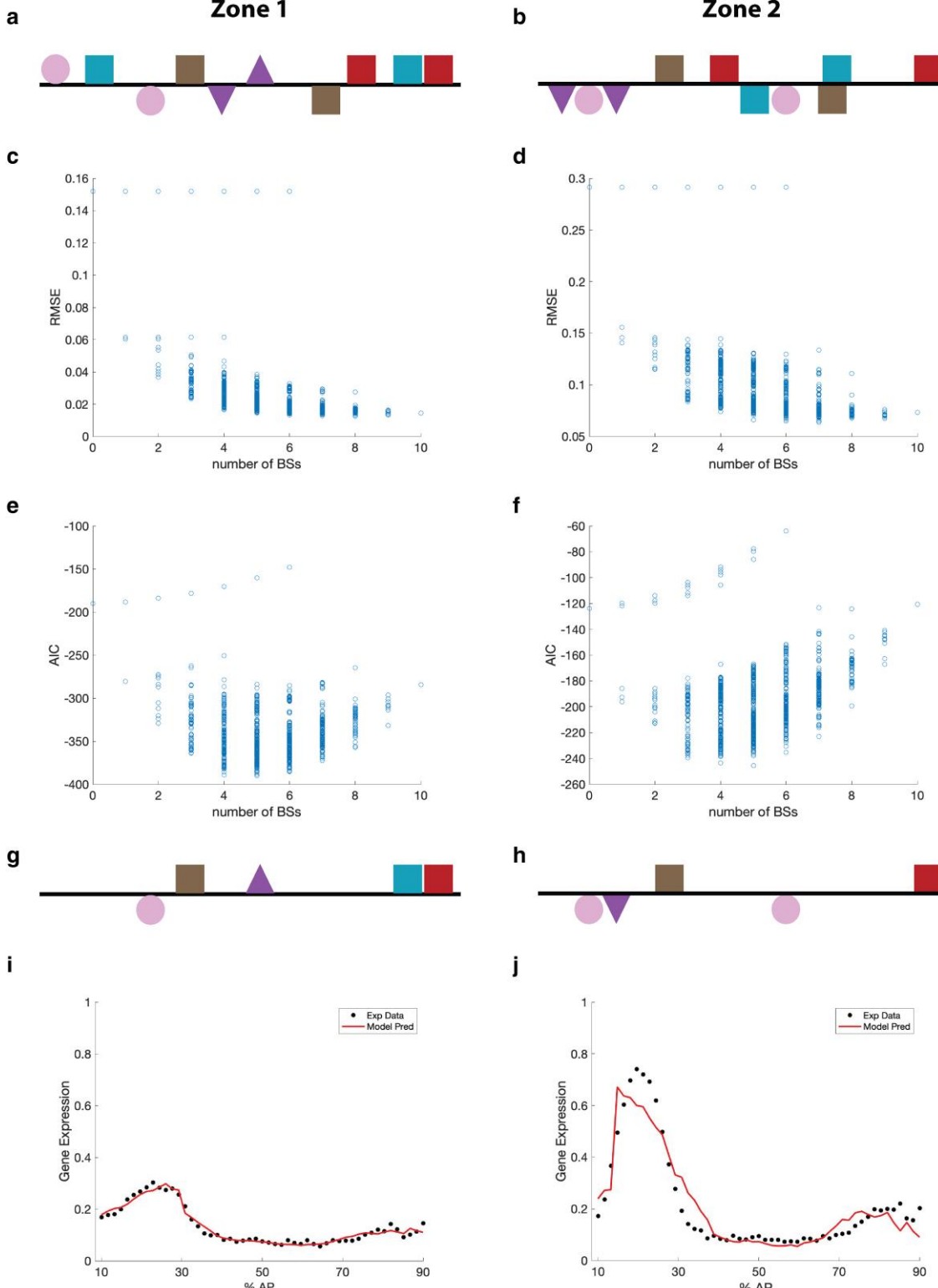

**Fig. 7.** Model performance on zones 1 and 2. Spatial arrangement of the 10 Transcription Factor binding sites (TFBSs) predicted by a PWM-based algorithm in zone 1 (a) and zone 2 (b) that were used to construct 1,024 thermodynamic-based models for each of the two enhancers. For each enhancer, the two strongest BCD (pink), CAD (blue), HB (purple), KNI (brown) and KR (red) binding sites are selected. The minimum root mean square error (RMSE) found for each of the 1,024 models when fit to the experimental expression data for zone 1 (c) and 2 (d) as a function of the number of binding sites (BSs) included in each model tested. The Akaike Information Criterion (AIC) value calculated from the minimum RMSE for each of the 1,024 models for zone 1 (e) and 2 (f) as a function of the number of BSs included in each model tested. The binding sites included in the best-performing models, as measured by AIC, for zone 1 (g) and 2 (h) are shown, along with the model predictions (red line) and experimental expression data (black dots) for zone 1 (i) and 2 (j). The x-axis represents the position along the anterior–posterior (AP) axis from 10% to 90%, and the y-axis represents normalized expression level.

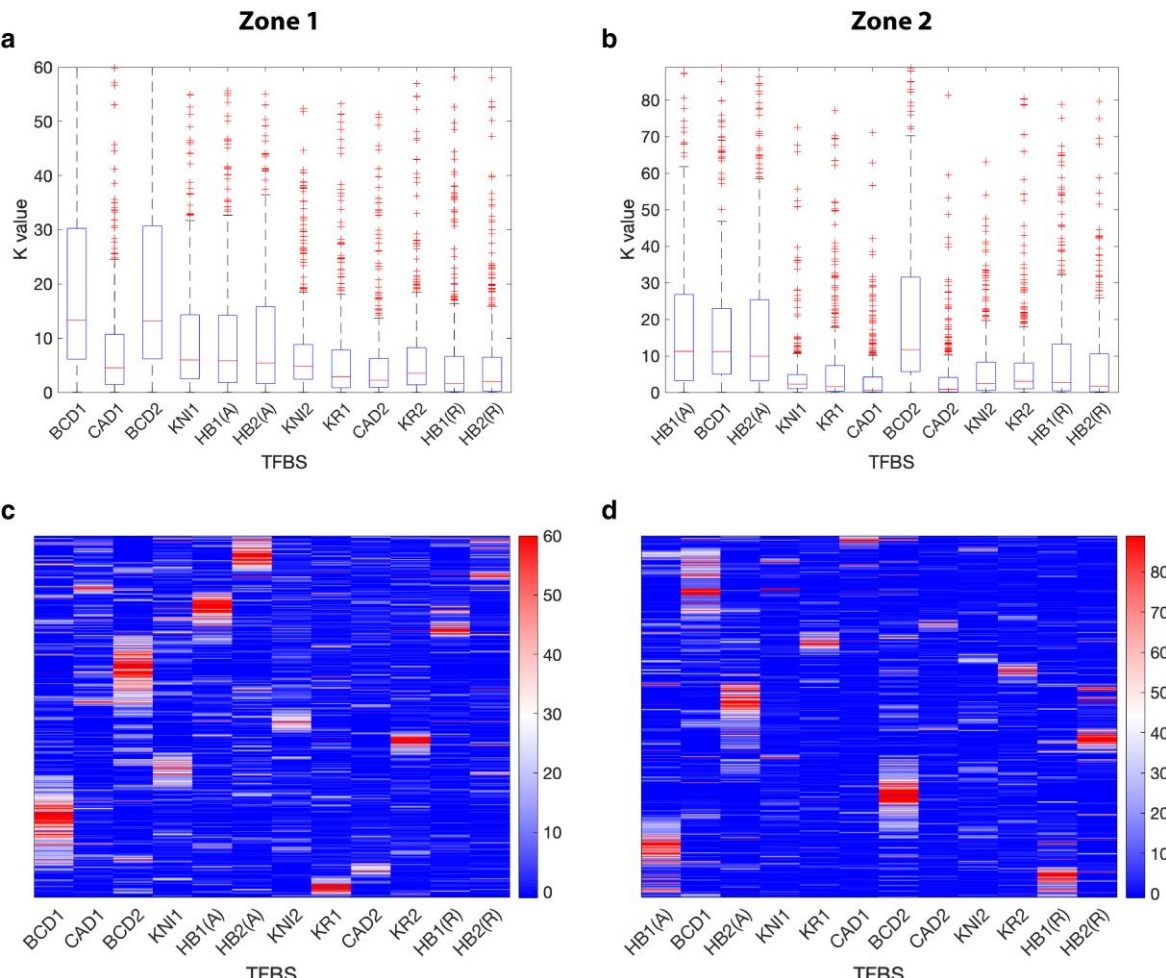

**Figure 8.** TFBS scaling parameters from all 2,048 models tested. Box-and-whisker plot of *K*-values obtained for each predicted transcription factor binding site (TFBS) included in the 1,024 models for each of the 2 enhancers, zone 1 (a) and zone 2 (b). The central mark indicates the median, the box indicates values falling between the 25th and 75th percentiles, the whiskers include all data points not considered outliers, and the outliers are plotted as red crosses. The upper limits on the y-axes were set to include at least 95% of all *K*-values (values above this are not shown). Note that models that do not include a specific TFBS did not include a *K*-value for that binding site, thus each *K*-value was fit for 512 models. Labels on the x-axes correspond to predicted binding sites for BCD, CAD, HB, KNI, and KR. To investigate the potential dual activity of HB in models that include at least one HB binding site, two parameters were fit for each binding site to account for activation (i.e. HB1(A)) and repression (i.e. HB1(R)) activity. Heatmaps depict the *K*-values for the TFBSs in all 1,024 models analyzed for zone 1 (c) and 2 (d). Each row of the heatmap corresponds to a single model. Thus, models that do not include a specific TFBS did not include a *K*-value for that binding site, which is equivalent to setting that *K*-value equal to zero. Therefore, these models are shown to have a *K*-value of 0 for the TFBS not included.

the HB2 site. Recall that these Q values are interpreted as the proportion of the state that contributes to activation when both TFs are bound; thus, these small values represent strong repression in these scenarios. In summary, the results suggest that the activation at the zone 1 enhancer is primarily driven by the BCD and HB activators acting independently, while the repressive activity on BCD appears to be primarily impacted by repressive cooperativity between KNI, CAD, and KR, and the only important repressive activity on HB appears to come from KNI.

For the zone 2 enhancer, the best-performing model (measured by AIC) resulted in an RMSE of approximately 0.07 when compared to the experimental expression data (Fig. 7, h and j). The parameter values resulting in this RMSE contained a very large *K*-value (>100) for the HB2 activator site and low *K*-values ($1 < K < 3.5$) for the two KNI1 and KR2 repressor sites (Supplementary Table 2). In contrast to zone 1, two BCD sites are also present in the best-performing model for zone 2 (Fig. 7h). Notably, the C values representing cooperativity between the BCD1 and BCD2 activator sites, along with KNI1-KR2 repressor cooperativity,

were found to be amongst the largest of any cooperativity values ($C > 50$) in the dataset. Moderate cooperativity was also reported in the case of the BCD1-HB2 activator interaction ($2 < C < 10$). Quenching values in the model were relatively small ($Q < 0.4$), representing strong repression, for KR2-BCD1, KNI1-HB2, and KR2-HB2. Taken together, the results suggest that the regulation of the zone 2 enhancer is more complex than that of the zone 1 enhancer, with a potentially important role for multiple TF interactions including cooperative interactions between the BCD sites and also repressor pairs containing KR. However, the fact remains that a relatively simple model with only 5 binding sites still resulted in the best fit to the experimental expression data for the zone 2 enhancer.

To further test the performance of the models, we investigated the impact of the reassortment of the TFBSs in zones 1 and 2 by randomly rearranging the five binding sites represented in the best-performing models for each zone 100 times and refitting the thermodynamic-based model (Supplementary Fig. 4, a–d). For each zone, not a single rearrangement resulted in an RMSE

below that obtained from the actual binding sites (Supplementary Fig. 4, e and f), indicating that the arrangement of the binding sites in zone 1 and 2 enhancers is optimal in terms of model performance. Intriguingly, our models also indicated a potential dual role for HB as both an activator and repressor for expression driven by the enhancers in different regions of the embryo.

## Dual-regulatory modality of HUNCHBACK

To address whether HB exhibits dual-regulatory activity when bound to either of the enhancers, we incorporated additional parameters in the thermodynamic-based models, including a threshold parameter which defines the position (along the AP axis) where HB switches from activating to repressing. The logic for doing so is provided by evidence from previous studies that HB has a concentration-dependent dual function in certain regulatory contexts (Papatsenko and Levine 2008; Ilsley *et al.* 2013; Staller *et al.* 2015). Paradoxically, the original Gehring lab studies on the genetic control of *toy* expression classified HB as a repressor, despite the relatively high concentration of HB in the anterior region of the embryo in which *toy* is expressed at stage 5 of development (Blanco and Gehring 2008). We therefore tested HB as an activator (i.e. HB1(A)) and repressor (i.e. HB1(R)) in our models. Overall, the HB repressor sites are found to have very low (<5) median $K$-values in both enhancer zones (Fig. 8, a and b), but there are distinct clusters of models in which their $K$-values are much higher (>40) (Fig. 8, c and d).

To directly test the spatial arrangement of the dual modality of HB in the embryo, we examined the switching of the function from activator to repressor across the AP axis within the models. Strikingly, very clear patterns emerged for the zone 1 and zone 2 enhancers when we analyzed the switching threshold value across all 768 models that included at least one HB binding site. For zone 1, the majority of threshold values (537) were found to be at nuclei 4, 14, or 50 (corresponding to 15, 31, and 88% AP) with the largest frequency (221) occurring at the 14th nucleus (31% AP), although many different threshold values were found amongst the different models (Fig. 9a). In contrast, for zone 2, the threshold values were all found in the range of 3–14 or 50–51 nuclei (corresponding to 13–31 and 88–90% AP) with the majority of models (430) reporting a single threshold value at the 11th nucleus (26% AP) (Fig. 9b). To address the possibility that HB may have the inverse functionality, we also examined the switching of the function from repressor to activator across the AP axis within the models. Again, a very clear pattern emerged for both enhancers (Fig. 9, c and d). For zone 1, the majority of threshold values (459) were found to be at nuclei 1–4 (corresponding to 10–15% AP) with the largest frequency (210) occurring at the fourth nucleus (15% AP) (Fig. 9c). For zone 2, the majority of threshold values (547) were found to be at nuclei 1–3 (corresponding to 10–13% AP) with the largest frequency (315) occurring at the 3rd nucleus (13% AP) (Fig. 9d). These threshold values point to a true dual modality for HB in regulating spatial expression from both enhancers, with its repressive activity playing a crucial role in setting the boundaries of *toy* expression while its function as an activator appears to be indispensable to allow for proper activation in the peak expression region (Fig. 9e, see Discussion for details).

## Discussion
### Identification of enhancers and epigenetic environment at toy

In this study, we adopted an integrated approach utilizing bioinformatics, molecular genetics, and mathematical modeling to identify early embryonic enhancers of *toy*, test the quantitative

transcriptional output of the enhancers, and dissect the regulatory logic controlling their activity. We initially identified 16 genomic zones in and around the *toy* gene sharing the hallmarks of putative regulatory elements, namely sequence conservation in different *Drosophila* species, an accessible chromatin environment and evidence of TF binding in vivo (Figs. 1 and 2). More detailed analysis of these characteristics enabled us to discover that 4 of the zones (1, 2, 4, and 7), all located upstream (5′) of the *toy* transcription start site, appeared to be enriched for these features (Fig. 3). Molecular testing of the 4 putative enhancers in transgenic reporter gene assays revealed that 2 zones (1 and 2) demonstrate orientation-independent regulatory activity in embryos, while 2 zones (4 and 7) have no activity (Fig. 4), representing a 50% predictive success rate.

To address if there are additional features of the four zones that correlate with their enhancer activity, we investigated the epigenetic profile at the genomic region around *toy* (Supplementary Fig. 3). Notably, all four zones fall in the same topologically associating domain (TAD) (Kerpedjiev *et al.* 2018), which appears to span from zone 1 to zone 7. The boundaries of the TAD are marked by a co-localized CTCF, ATAC-seq, and ZELDA-negative peak in the center of zone 1 and an ATAC-seq and ZELDA-negative peak in zone 7 in the promoter region of the *toy* gene (Supplementary Fig. 3). In addition, zones 1 and 2 demonstrate an enrichment of binding sites for the ZELDA pioneer TF, when compared to zones 4 and 7 (Supplementary Table 3). None of the four zones exhibit enrichment of the active enhancer epigenetic mark, H3K27 acetylation, when compared to neighboring genomic regions, which may account for the fact that the enhancer atlas database (Gao and Qian 2020) mistakenly only identifies zone 4 as an early embryonic enhancer. It should also be noted that the *lacZ* reporter constructs in our study utilize a basal *hsp70* promoter and not the endogenous *toy* promoter region. As a result, our in vivo analysis in embryos does not investigate the potential for specific enhancer–promoter interactions at the *toy* locus, as have been characterized for other genes expressed in early *Drosophila* development (Calhoun *et al.* 2002; Akbari *et al.* 2008).

### Distinct regulatory logic at toy enhancers

Comparison of the reporter gene expression profiles from the zone 1 and 2 enhancers revealed similar, but not identical, patterns in the same locations as previously reported endogenous *toy* expression (Figs. 4 and 5; Czerny *et al.* 1999). Quantitative analysis showed that zone 1 drives an overall lower level of expression when compared to zone 2, and that the domain of anterior expression for zone 1 is broader (Fig. 6). To further explore the molecular organization of the two enhancers we examined the evolutionary conservation of predicted Transcription Factor binding sites (TFBSs) across *Drosophila* species. Strikingly, the underlying regulatory logic appears to be different between the two enhancers (Fig. 10). The zone 1 enhancer demonstrates a widely dispersed spatial organization of the identified TFBSs and weak conservation, even in closely related species (Fig. 10a). In contrast, 9 out of the 10 TFBSs in the zone 2 enhancer are tightly clustered and largely conserved, even in distantly related species (Fig. 10b). The conserved cluster of binding sites in zone 2 reveals an evolutionary signature motif, consisting of two BCD sites and at least one site for each of CAD, HB, KNI and KR. If we consider the prevailing models of enhancer architecture (Borok *et al.* 2010; Panigrahi and O'Malley 2021), zone 1 appears to demonstrate a clear "billboard" organization, that is a collection of TFBSs that function with a high degree of independence from each other. This enhancer architecture confers low conservation in the spatial organization of the TFBSs and supports additive recruitment

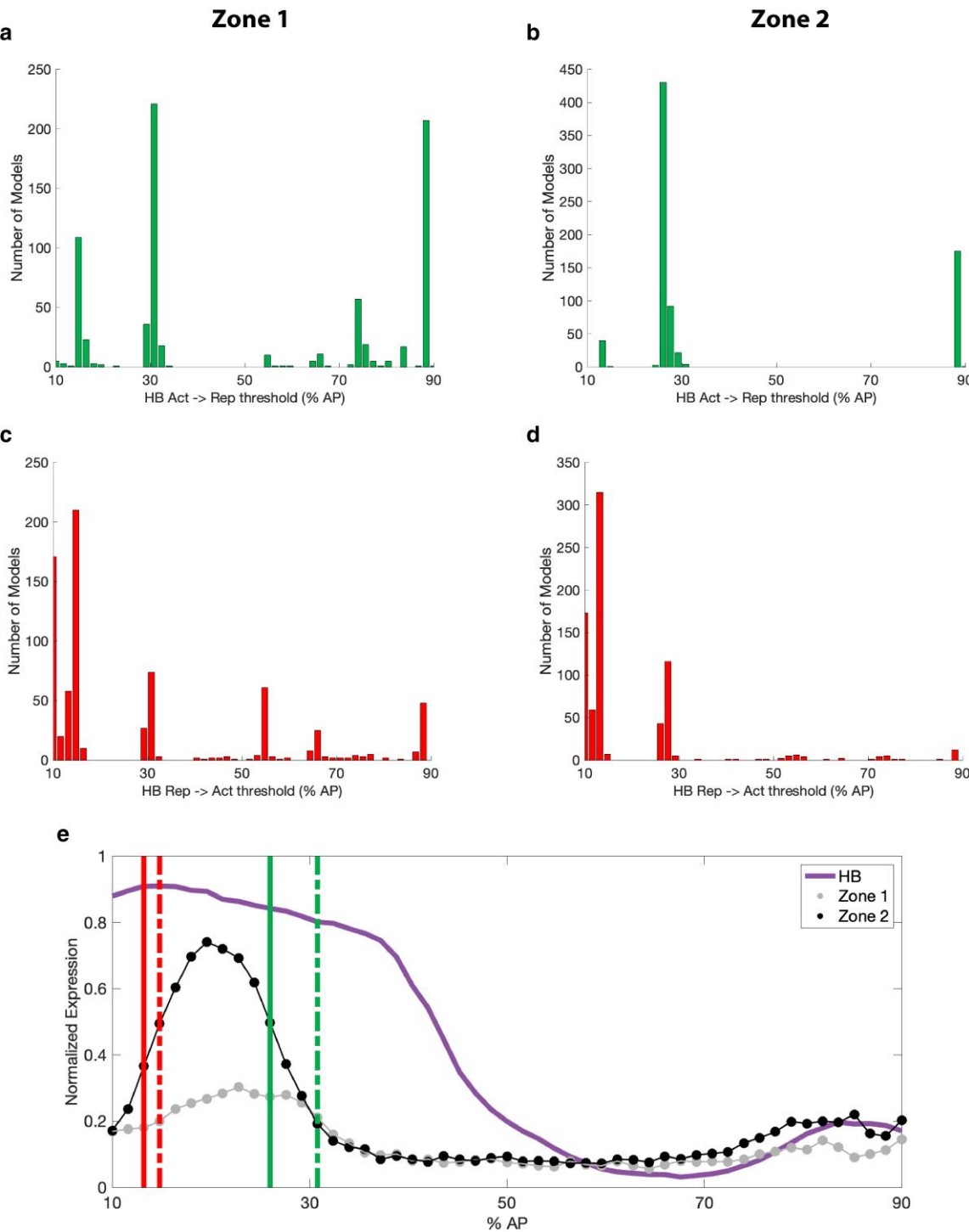

**Fig. 9.** HB functional duality thresholds. Histogram of HB threshold values found when models were fit to experimental data for each of the two enhancers. The threshold value represents the nucleus on the anterior–posterior (AP) axis in which the model assumes the regulatory role for HB activity switches from activation to repression from zone 1 (a) and 2 (b), or repression to activation from zone 1 (c) and 2 (d). In each histogram, the x-axis corresponds to the threshold value (% AP) approximated during model fitting and the y-axis corresponds to the number of models that resulted in that exact threshold value. Note that this data comes from a total of 768 models, all of which include at least one of the two predicted HB binding sites used for modeling. e) Quantitative analysis of HB duality in embryo. Concentration of the HB protein (purple) from the BDTNP dataset (Fowlkes et al. 2008) is shown 80 minutes into development, along with experimental reporter gene expression driven by zone 1 (gray) and zone 2 (black) enhancers. The red (repressor to activator switch) and green (activator to repressor switch) lines correspond to the location of the threshold value (%AP) found with the highest frequency among the 768 models including at least one HB binding site. Dotted lines correspond to values obtained from modeling zone 1 and solid lines correspond to values obtained from modeling zone 2.

of the TFs (Arnosti and Kulkarni 2005; Borok et al. 2010; Long et al. 2016). On the contrary, the "enhanceosome" model, which seems to fit the signature motif organization of the zone 2 enhancer, proposes that TFs bind cooperatively within one functional unit (Merika and Thanos 2001; Arnosti and Kulkarni 2005; Borok et al. 2010). Accordingly, extensive alterations in the sequence, order,

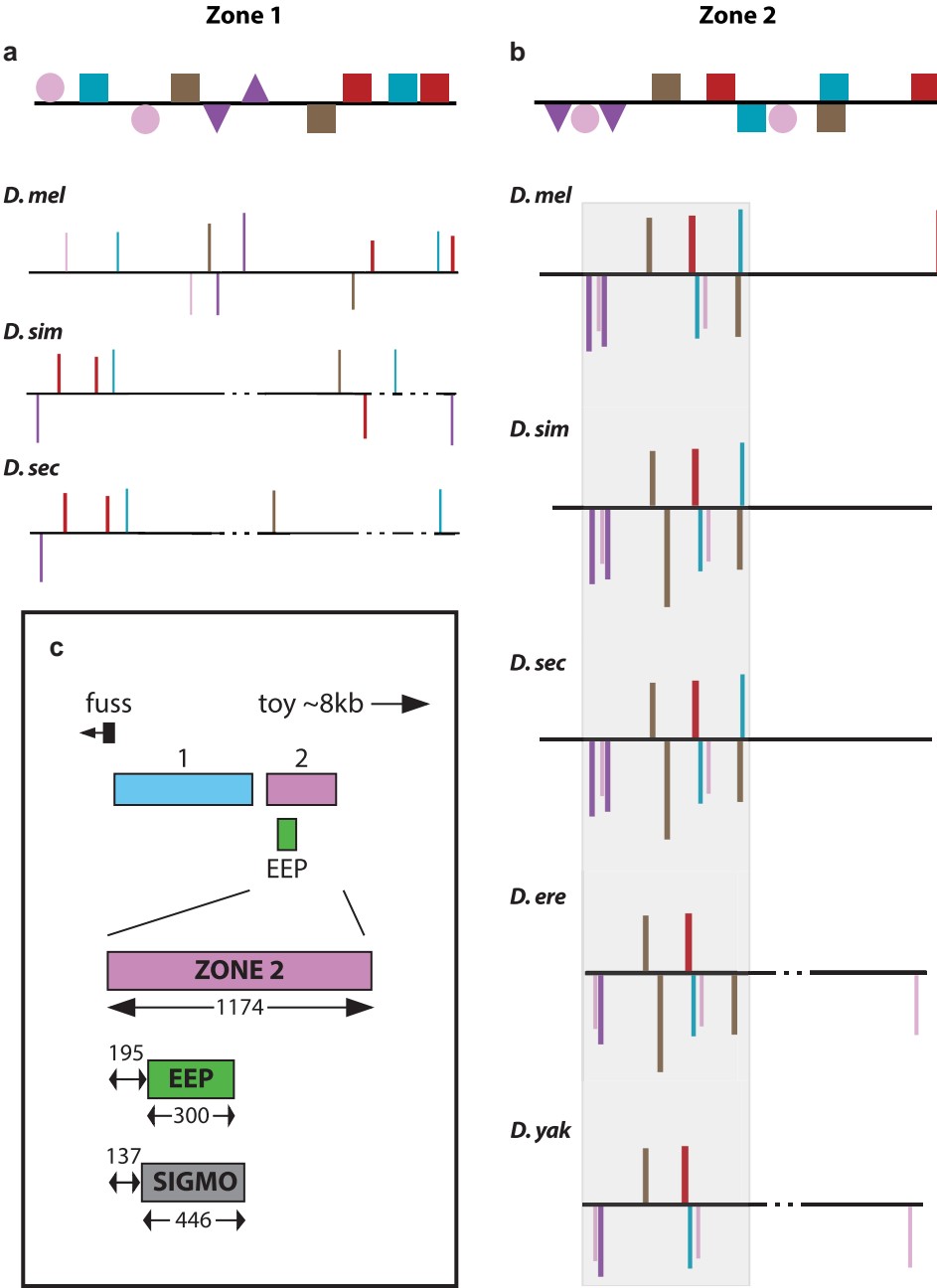

**Figure 10.** Evolutionary conservation of TFBSs at zone 1 and 2 enhancers across *Drosophila* species. Spatial arrangement of the 10 Transcription Factor binding sites (TFBSs) predicted by a PWM-based algorithm in the zone 1 enhancer (a) and the zone 2 enhancer (b) that were used to construct the thermodynamic-based models for each of the 2 enhancers. For each enhancer, the two strongest BCD activator (pink circle), CAD repressor (blue square), HB dual-regulatory function (purple triangle), KNI repressor (brown square) and KR repressor (red square) binding sites are shown. For individual species, the height of bars indicates the similarity of the given site to the binding sites used to construct the PWM, the top or bottom of the bar indicates on which strand the predicted binding site is found and dashed lines indicate a gap in the sequence alignment. The cluster of TFBSs in zone 2 reveals a conserved signature motif, consisting of two BCD sites and at least one site for each of CAD, HB, KNI, and KR (gray box). (c) The organization of the *toy* early embryonic enhancers. The signature motif (SIGMO) is located in the 5′ half of zone 2 and encompasses the previously identified EEP.

or spacing of the TFBSs are not tolerated, and these enhancers represent a class that is under high evolutionary constraint and thus highly conserved in different species (Arnosti and Kulkarni 2005; Crocker and Erives 2008; Hare *et al.* 2008; Ho *et al.* 2009; Ludwig *et al.* 2011; Starr *et al.* 2011; Drewell *et al.* 2014). Neither the billboard (zone 1) nor enhanceosome (zone 2) models preclude the potential contribution of a third model of regulatory logic organization at enhancers, known as the "TF collective" model. In this model, TFs can be recruited to the enhancer not only through direct binding with their respective binding sites but also through

protein–protein interactions, which may also help recruit additional TFs that play a role in regulation (Spitz and Furlong 2012; Khoueiry *et al.* 2017). Further experimental testing will be required to dissect the functional organization of the TFBSs in each enhancer and specifically address whether the conserved spatial organization found in zone 2 is required for the proper *cis*-regulatory output. In addition, generating targeted mutations of the key TFBSs identified in our bioinformatic and modeling studies and testing those in reporter gene assays will be critical to confirm their role in enhancer activity.

In an effort to further test the performance of our models of enhancer function, we investigated two additional well-characterized enhancers known to be regulated by BICOID; even-skipped stripe 2 (eve2) (Arnosti, Barolo *et al.* 1996) and giant 23 (gt23) (Ochoa-Espinosa *et al.* 2005). For eve2, we modeled the 12 TFBSs that were previously identified (Arnosti, Barolo *et al.* 1996), including sites for BCD, HB, KR, and GT. For gt23, we used a PWM-based algorithm to predict 2 binding sites for each known TF regulator; BCD, HB, and KR (Ochoa-Espinosa *et al.* 2005). In both cases, our thermodynamic-based model performs well and largely recapitulates the previously characterized expression pattern driven by these enhancers (Supplementary Fig. 5). These proof-of-principle results suggest there may be some shared regulatory logic at play in BICOID-driven gene expression and open the door to larger-scale modeling experiments to address the underlying regulatory rules within this network of TFs in the early embryo.

## Shadow enhancers

Given their genomic location adjacent to the *toy* gene and overlapping spatiotemporal expression patterns, zones 1 and 2 exhibit the defining characteristics of shadow enhancers (Hong *et al.* 2008; Barolo 2012; Waymack *et al.* 2020). Such coacting enhancer systems are a pervasive feature of gene regulatory networks (Cannavò *et al.* 2016), including the regulatory cascade that controls early *Drosophila* development (Perry *et al.* 2010; Wunderlich *et al.* 2015; El-Sherif and Levine 2016; Scholes *et al.* 2019). Prior studies have suggested that shadow enhancers evolved as a mechanism to achieve 3 interconnected regulatory objectives, to confer transcriptional robustness (Perry *et al.* 2010; Swami 2010; Fujiwara and Cañestro 2018), to integrate input TF signals (Scholes *et al.* 2019), and to mediate tradeoffs between transcriptional noise and fidelity (Waymack *et al.* 2020; Fletcher *et al.* 2023). As the precise spatiotemporal expression of *toy* at the very pinnacle of the retinal determination network is critical to eye development in *Drosophila*, it is perhaps not surprising to find that two distinct enhancers are responsible for directing expression in the early embryo. It is also possible that the two enhancers may be capable of cross-regulating each other, as has been demonstrated for the shadow enhancers at the *hunchback* and *knirps* genes (Perry *et al.* 2011). To address this possibility, we modeled the output from the concatenated *toy* enhancers, in which adjacent TFBSs can interact with each other (see Methods for details). Intriguingly, the predicted pattern of expression from the 2 combined enhancers more closely resembles endogenous *toy* expression (Karaiskos *et al.* 2017), when compared to the predicted pattern driven by zone 1 or zone 2 alone (Supplementary Fig. 4g). Further molecular studies will be essential to dissect the detailed functional contribution of the coacting enhancers, and the associated regulatory objectives, to *toy* expression. However, it should be noted that in this case, the two enhancers appear to be employing very different internal regulatory architectures to mediate extensively overlapping expression patterns, as detailed above. The discovery of the conserved TFBS signature motif in the zone 2 enhancer, and the fact that the previously identified embryonic eye primordium-specific (EEP) enhancer (Blanco *et al.* 2010) is contained within it (Fig. 10c), indicates that this minimal regulatory region contributes significantly to endogenous *toy* expression.

## Complex regulatory landscape includes HUNCHBACK dual functional activity

Mathematical modeling of the control of the transcriptional output from the two enhancers revealed that relatively simple thermodynamic-based models that incorporate just five TFBSs can accurately predict expression from the zone 1 and zone 2 enhancers (Fig. 7). Furthermore, the models allow us to systematically investigate the molecular interactions taking place at each enhancer (Fig. 8) and classify the functional activity of the key TFs predicted to bind at the enhancers. Accordingly, BCD appears to act solely as an activator, while CAD, KNI, and KR are exclusively repressors at both coacting enhancers. These modeling results support the interpretation of the role of these TFs from earlier genetic studies in the Gehring lab (Blanco and Gehring 2008). However, our models also uncover a spatially modulated dual function for HB as a potential activator and repressor of *toy* expression (Fig. 9). Prior studies have identified this HB dual-regulatory modality in *Drosophila* embryonic development (Papatsenko and Levine 2008) and there is also evidence that shadow enhancers may provide a mechanistic opportunity to enable this bifunctionality (Staller *et al.* 2015).

Analysis of the thermodynamic-based models of zone 1 and 2 activity allowed us to identify spatial thresholds along the anterior–posterior (AP) axis where HB switches from an activator to a repressor (or vice-versa). Connecting these critical threshold values with the quantitative gene expression output from the enhancers, one should recall that the peak expression for zones 1 and 2 occurs at 26.04% AP and 21.30% AP, respectively. It is therefore reassuring that we find the highest frequency of threshold values for HB switching from an activator to a repressor at 32% AP for zone 1 and 27% AP for zone 2, just beyond the posterior boundary of the expression peaks (Fig. 9e). Furthermore, the highest frequency of threshold values for the inverse HB switching, from a repressor to an activator, is at 15% AP for zone 1 and 13% AP for zone 2; just prior to the anterior boundary of the expression peaks (Fig. 9e). In summary, these results suggest that the role of HB as a repressor for *toy* reported in earlier studies (Blanco and Gehring 2008) may be accurate only for regions of the embryo with the very highest levels of HB (in the most anterior part of the embryo) and/or in regions where there is a co-repressor present to modify its function (Fig. 9e), but classifying HB solely as a repressor of *toy* is an oversimplification. Rather, our results support a hypothesis that high concentrations of HB facilitate activation, but that there is a role for HB as a repressor in defining the boundaries of *toy* expression in the early embryo (Fig. 9e). In agreement with our model predictions, in HB mutant embryos *toy* is expressed slightly earlier and expands toward the anterior (Blanco and Gehring 2008). In addition, we note that there is a small, but reproducibly measurable region of expression directed by both the zone 1 and zone 2 enhancers in the posterior of the embryo (Figs. 4 and 5) that aligns with the corresponding minor peak of HB protein present in this posterior region at stage 5 of embryonic development (Fig. 9e). To gain a more holistic view of HB activity, one must consider the possibility that the TF collective model may be at play here, with HB recruited to the two coacting enhancers using different mechanisms and/or interactions with the other regulatory TFs bound, using distinct regulatory logic at each enhancer. Further investigation into how this might be controlled at the molecular level, including investigating the concentration dependence and synergistic repression with CAD, KNI, KR, or other co-repressors not currently included in our modeling efforts, will be an important avenue to explore in future studies. Another addition to the modeling will be to include predicted weaker binding sites for the TFs we have found to be important for the expression directed by each enhancer, as there is a growing body of evidence that suboptimal TF binding sites can play a significant role in gene regulation (White *et al.* 2012; Kim, Rhee *et al.* 2022). To further validate

the modeling results, analyzing the expression of the *lacZ* reporter constructs in selected TF mutant backgrounds, including HB, in future studies would be very informative. A complete understanding of the regulatory activity of the two coacting enhancers will also require a molecular dissection of how they modulate *toy* expression in a spatially restricted pattern across the DV axis.

## Data availability

Strains and plasmids are available upon request. The authors affirm that all data necessary for confirming the conclusions of the article are present within the article, figures, and tables.

Supplemental material available at GENETICS online.

## Funding

This work was funded in part by National Institutes of Health grants (GM110571 and GM137250) to RAD and JMD.

## Conflicts of interest

The authors report no competing interests.

## Author contributions

JMD and RAD conceived of the study, participated in its design, established the analysis pipelines, coordinated and drafted the manuscript. LLN participated in the design of the study, carried out sequence and bioinformatic analysis and performed molecular cloning and in situ hybridization experiments. RDC developed the quantitative analysis and, along with LTC, EIT and EV, participated in the mathematical modeling in the study. BT performed in situ hybridization experiments. MG performed sequence conservation analysis. DB-R performed the epigenetic profile analysis. All authors read and approved the final manuscript.

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

*Editor: P. Geyer*
