## [Peer Review File · Genetics]

Two coacting shadow enhancers regulate twin of eyeless expression during early *Drosophila* development

Jacqueline Dresch, Luke Nourie, Regan Conrad, Lindsay Carlson, Elizabeth Tchantouridze, Biruck Tesfaye, Eleanor Verhagen, Mahima Gupta, Diego Borges-Rivera, and Robert Drewell

NOTE: The reviews and decision letters are unedited and appear as submitted by the reviewers.

In extremely rare instances and as determined by a Senior Editor or the EIC, portions of a review may be redacted. If a review is signed, the reviewer has agreed to no longer remain anonymous.

The review history appears in chronological order.

Review Timeline:

Submission Date:	2024-08-07
Editorial Decision:	2024-09-18
Resubmission Received:	2024-10-18
Accepted:	2024-10-21

September 18, 2024

GENETICS-2024-307356

Two coacting shadow enhancers regulate twin of eyeless expression during early Drosophila development

Dear Robert:

Three experts in the field have reviewed your manuscript, and I have read it as well. All agree that your manuscript includes an impressive combination of experimental approaches and have provided new insights into the transcriptional regulation of a key developmental gene. Based on their responses, I am pleased to inform you that, with minor revisions, your manuscript is potentially suitable for publication in GENETICS. The reviewers have comments and concerns that need to be addressed in a revised manuscript. You can read their reviews at the end of this email.

Although several new experiments were proposed, suggestions made concerning expanding the application of your trained model to test additional regulatory elements should be strongly considered. For example, can you model a regulatory region that combines zones 1 and 2? What outputs result from modeling of zones 4 and 7? Can your models be used to predict the regulatory output of other Bcd enhancers?

In addition to comments from the three reviewers, I would like you to address the following comments in your revision.

1. Please update formatting
 - a. remove subheadings in the introduction.
 - b. Include more labeling in figures for better reader access to the information. For example, in Fig. 4, instead of LacZ indicate it is zone 2-lacZ. Also, indicate the stages studied in the figure.
 - c. Please add size of each zone to table 1.
2. Please expand your description of K values and cooperativity value and provide more detailed information on data in Figure 7. In the introduction, you introduce the concept of high-order cooperativity, please define.

We look forward to receiving your revised manuscript. Please let the editorial office know approximately how long you expect to need for revisions.

Upon resubmission, please include:

1. A clean version of your manuscript;
2. A marked version of your manuscript in which you highlight significant revisions carried out in response to the major points raised by the editor/reviewers (track changes is acceptable if preferred);
3. A detailed response to the editor's/reviewers' comments and to the concerns listed above. Please reference line numbers in this response to aid the editors.

Additionally, please ensure that your resubmission is formatted for GENETICS.

<https://academic.oup.com/genetics/pages/general-instructions>

Follow this link to submit the revised manuscript: Link Not Available

Sincerely,

Pamela Geyer
Associate Editor
GENETICS

Approved by:
Karen Arndt
Senior Editor
GENETICS

Reviewer #1 :

toy is a key drosophila gene that controls eye development. Dresch et al. investigated how toy gene expression is regulated by enhancers and multiple TFs using both computational and experimental approaches. In the manuscript, authors selected four genomic zones as potential enhancer regions for toy gene based on bioinformatic analysis. Two of them (zone 1 and 2) were further validated by lacZ reporter assays, which showed that both enhancers can drive an expression pattern similar to the endogenous gene. Additionally, the authors implemented mathematical modeling to predict TF binding sites and TF functionality for toy gene regulation, revealing distinct TF regulatory patterns for zone 1 and zone 2. Moreover, one of the TFs, HUNCHBACK, likely acts as both an activator and repressor for toy gene expression.

The manuscript is well drafted, with experimental and mathematical approaches documented in detail. The story is very interesting and has a significant impact on developmental biology and enhancer-driven gene regulation. However, quality of the manuscript could be improved with additional genetic investigation.

Major issues/questions:

1. The two zones exhibit distinct regulatory landscape by TFs. What pattern of toy gene expression would be observed if the reporter is driven by both zone 1 and zone 2? Given these two zones are adjacent in the genome. Would concatenated enhancers behave more like zone 2, similar to toy expression?
2. To further validate the modeling results, lacZ reporter assays in mutants of select TFs should be conducted. While testing every TF is unnecessary, focusing on several important ones, such as HUNCHBACK, would be valuable.

Minor issues:

1. In Figure 3, the embryos in the bottom two rows of panels (zones 4 and 7) appear noticeably smaller than those in the top two rows. Do the authors present the representative images, or is there an additional explanation for the size discrepancy?
2. A diagram to summarize the main findings is better to be included to help the readers better understand the story.

Reviewer #2 :

In this paper, the authors examine the regulatory landscape surrounding the twin of eyeless (toy) locus. Specifically, they describe two new embryonic enhancers driving expression in stage 5 embryos. They accomplish this by bioinformatically predicting enhancer regions and generating LacZ reporter constructs driven by the putative enhancer elements. Once they identified two putative enhancer elements, they utilized thermodynamic mathematical models to predict transcription factor binding of known regulators of toy as well as determine their roles as activators or repressors.

Overall, this paper is excellent and offers new insights into the regulation of an important transcription factor that is vital for nervous system and eye development in Drosophila. The data presented in this manuscript support the author's conclusions that these regulatory regions drive expression of LacZ in the pattern of toy during development, however, some additional experiments or modifications to language would be helpful in matching the experimental dataset with the conclusions of the paper.

Major comments:

1. Expand time course of enhancer experiments: It is unclear how you decided to focus on stage 5 (other than this is the onset of toy expression in the embryo). Does the expression drive by these enhancers diverge later in embryonic development? As they already exhibit differences in pattern of expression, conducting the LacZ in situ at later embryonic stages would offer a stronger argument that these enhancers have largely overlapping expression domains.
2. Determine functional role of binding sites within the LacZ reporter lines: The mathematic model identifies important binding sites as activators and repressors of expression at the Zone 1 and Zone 2 putative enhancers. As proof of principle, generating mutations at select binding sites within the LacZ construct would offer a significantly stronger argument for their role in enhancer activity.

Minor comments:

1. Examine chromatin accessibility at zone 1 and zone 2 regions at other time points: It's interesting that the chromatin at the Zone 1 and Zone two putative enhancers don't have H3K27ac. As chromatin state plays a major role in spatial and temporal regulation of gene expression, it would be helpful to examine available ChIP-seq datasets to identify if chromatin is marked with H3K27ac at other points in development.

Other comments/suggestions:

1. Did you look for other TFBS other than the known regulators? Are there significantly different predicted TFBS between Zone 1 and Zone 2? This also may be helpful in trying to determine if these are acting as shadow enhancers.
2. Line 140, the figure call for Fig 2A, stylistically, Figure 1 may be a better representation of this statement.

Reviewer #3 :

From the first experiments that examined the combined action of transcription factors on enhancers, it is clear that the activity of activators and repressors depends on a complex regulatory "grammar" that is only incompletely understood. With an increasing molecular description of enhancers in their native state in vivo, we have larger data sets with which to test models for how enhancers influence gene expression. Thermodynamic modeling, pioneered by the Reinitz laboratory for studies of embryonic gene expression in *Drosophila*, uses physical properties of transcription factor-DNA interactions and models their interactions with DNA, each other, and the transcriptional machinery to predict gene expression. Dresch, Drewell and colleagues have employed this approach successfully in the past to identify possible mechanisms of gene regulation in *Drosophila*. In this study, Dresch et al. identify a new element involved in regulation of the important toy gene, and employ thermodynamic modeling to predict how Bicoid, Hunchback, Knirps, Kruppel, and Caudal regulate the gene from two individual regions. The modeling indicates that the internal logic of regions 1 and 2 have some differences in how activators and repressors interact, such as differences in the amount of cooperativity, the relative importance of individual binding sites, and interestingly, the action of Hunchback as either a repressor or an activator. The authors explicitly test the idea that Hunchback may act differentially in different spacial domains of the embryo, and show that models with specific activator/repressor switch positions are favored by many of the tested models. In sum, this enhancer characterization effort goes far beyond the traditional enhancer-identification approach, and proposes a highly context-dependent function for TF on closely linked elements that generate similar outputs.

The authors are invited to consider the following points:

- 1) The logic by which four regions of the toy gene are selected for experimental investigation is clearly spelled out, and there are assumptions that are untested, such as the relative importance of certain features in predicting active enhancers. As they note in the introduction, there are many predicted regulatory element in the toy locus. It would be revealing for the authors to apply some of their models (both best-fit and some mediocre?) to blocks of DNA throughout the toy locus, which as they show in the Supp Figs. 2 and 3 is replete with TF binding sites and accessible DNA. If some of the models predict outputs of a toy-like pattern, it is likely that these are additional redundant elements.
- 2) This exercise would be less interesting if no other portions of the toy locus have a toy-like output, as it might be a true or false negative. To further test the power of these models, it would be revealing to reassort the sets of bindings sites in regions 1 and 2, and apply the models to ascertain whether site arrangement, in addition to bulk composition of binding sites, is driving model performance.
- 3) Small and colleagues have identified a number of Bicoid-driven anterior gene expression patterns, noting that anterior-posterior positioning is not a simple function of Bicoid binding site affinity. The models trained on toy regions can similarly be tested on several of these elements, to test if there are any overarching similarities in regulatory logic. The paper would then get at a much larger question of how individual are the regulatory rules that direct the control of sets of TF. To my knowledge, this question has only been deeply addressed for individual transcription factors, such as ER and NF- κ B control elements on simpler target elements.
- 4) Knirps is largely expressed in ventral regions in the anterior of the embryo - why would it contribute to regulation in the 25% region tested here?
- 5) The modeling here is restricted to individual regions, not region 1 - region 2 cross regulation. The authors should note that there are examples of such enhancer-enhancer cross regulation (Perry et al. 2011 PNAS).
- 6) The authors identify the TF proteins in all capital letters, which is not the usual nomenclature.
- 7) All experimental constructs use the pLacZattB plasmid, which apparently employs the hsp70 basal promoter, not the toy basal region. The authors should discuss how enhancer-promoter interactions may or may not be relevant to this study.
- 8) K values are shown for individual TF in Fig. 7. It is not clear what the difference is between A/B and C/D; they don't seem to show the same values.

Associate Editor Comments:

Dr. Robert A. Drewell
Biology Department
Clark University
950 Main Street
Worcester, MA 01610

18th October 2024

RE: GENETICS-2024-307356

Dear Dr. Geyer

Please find attached our revised manuscript – **Two coacting shadow enhancers regulate *twin of eyeless* expression during early *Drosophila* development**. The comments and suggestions from the three reviewers and yourself were very valuable in making significant improvements to the paper. Attached are the revised manuscript, a copy of the manuscript with tracked changes, and revised figures/tables as needed. Below is a detailed point-by-point response to the reviewer's comments.

I hope you will find the revised manuscript now suitable for publication in *Genetics*.

The revised manuscript have been seen and approved by all listed authors.

Yours sincerely

Robert A. Drewell

Editor comments

1. Subheadings removed from *Introduction* section, additional labels added to figures 3, 4, 7 and 8 for clarity and the size of the zones added to table 1.
2. Text was added to the *Materials and Methods* in the Thermodynamic-based modeling section to expand the description of K values (lines 634-638) and C values (lines 641-644). In addition, a typo in the legend for Fig. 7 was corrected and we have included additional text clarifying the heatmaps in panels C and D (lines 773-777). We also added text to the *Introduction* section (lines 108-112) to define higher-order cooperative interactions.

Reviewer #1

Major

1. This is an excellent question. We modeled the predicted output from the concatenated enhancers and found that the expression more closely resembles endogenous *toy* than either enhancer alone. This new result is described on page 21 (lines 488-493) and shown in Fig. S4g.
2. The suggestion to test the *lacZ* reporter constructs in selected TF mutant backgrounds is another great idea. However, we feel that these experiments lie beyond the scope of the current study and therefore would be best performed in future studies. However, this idea is now incorporated into the *Discussion* section on page 23 (lines 547-549).

Minor

1. The embryos shown in Figure 3 are all shown to scale and are representative. Any size differences are simply due to the normal biological variation.
2. Figure 9 (and panel c in particular) showing the internal architecture of the two newly discovered enhancers and their organization at the *toy* locus is intended as a summary of the major findings of the paper.

Reviewer #2

Major

1. In Figure 4, we present a developmental time course of expression of zone 2-driven *lacZ* expression and compare it with the endogenous *toy* expression pattern. The zone 1 expression pattern is qualitatively indistinguishable from zone 2. This is now made clear in the text of *Results* section on page 9 (lines 191-194).
2. This is a great point. Generating mutations in the key TF binding sites identified in our bioinformatic and modeling studies and testing those in reporter gene assays will be critical to confirm their role in enhancer activity. We now mention this point in the *Discussion* section on page 20 (lines 451-453) as a key avenue to explore in future studies.

Minor

1. The BDTNP data presented in Figure 2c indicate that the chromatin at zone 1 remains accessible through to stage 14 of development. The accessibility at zone 2 appears to be more dynamic, with diminished accessibility beyond stage 10 of development. The absence of H3K27ac marks at zone 1 and zone 2 is confirmed in the *embryo 6-8h ChIP-seq His3 modifications* track in FlyBase JBrowse. We agree that a comprehensive exploration of the epigenetic and nuclear organization of the *toy* genomic regulatory region would be an exciting future research opportunity.

Other comments

1. For this study, we restricted our investigation of the TF binding sites to the previously identified regulators of *toy* expression. For these TFs, the total binding site density is approximately equal in zone 1 (23.27) and zone 2 (21.43) and there doesn't appear to be a striking enrichment for any of the eight TFs – see SI Table 1 for summary. We have now also added analysis of ZELDA binding sites - please see response to Reviewer 3, comment #1.

2. Figure call changed from Fig. 2a to Fig. 1.

Reviewer #3

1 and 2. These are great suggestions. We were surprised to find that zones 4 and 7 failed to demonstrate any enhancer activity in the *lacZ* reporter gene assay, despite that fact that they share many features with zone 1 and 2. We have now incorporated an additional analysis of predicted ZELDA binding sites (page 18, lines 415-416 and Table S3) that demonstrates an enrichment of binding sites for this 'pioneer' TF in zone 1 and 2, when compared to zone 4 and 7. In future studies there would be real value in applying an integrated approach of testing the models and the *in vivo* activity of the additional zones located 5' of the *toy* gene (zones 3, 5 and 6) alongside the interspersed blocks of genomic DNA. To investigate the impact of the reassortment of the TF binding sites in zone 1 and 2, we randomly rearranged the five binding sites in the best performing models for each zone 100 times and refit the thermodynamic-based model. For each zone, not a single rearrangement resulted in an RMSE below that obtained from the actual binding sites (Fig. S4e and f), indicating that the arrangement of the binding sites in the zone 1 and 2 enhancers is optimal in terms of model performance. These new results are now discussed on page 16 (lines 351-356) and presented in Fig. S4a-f.

3. We now test our thermodynamic-based model on two additional well characterized BICOID-regulated enhancers; even-skipped stripe 2 and giant 23. In both cases, the model performs well and recapitulates the previously characterized expression driven by these enhancers. These proof-of-principle results suggest there may be some shared regulatory logic at play and open the door to larger scale experiments to address the underlying regulatory rules within this network of TFs. These new results are presented on page 20 (lines 455-463) and in Fig. S5.

4. While *knirps* expression is predominantly restricted to the ventral side in the anterior region of the early embryo, there is a lower-level band of expression that extends across the DV axis in cellularizing embryos (see Fig. 3c in PMC1868043). In addition, in the more central region of the embryo, *knirps* expression is not restricted to the ventral side. *In vivo*, it is therefore likely that KNIRPS is functioning as a repressor of early *toy* expression. This conclusion is supported by the expansion of *toy* expression observed in the early Blanco and Gehring mutant studies (Reference #11 in paper).

5. Cross-enhancer regulation now noted in paper (page 21, lines 488-493) – see also reviewer 1, major comment #1.

6. We initially adopted the convention of using all caps to signify proteins in our publications beginning in the early 2010s, so it makes logical sense to continue with this naming scheme in the current paper.
7. Discussion of enhancer-promoter interactions added on page 19 (lines 419-422).
8. Please see editor comment #2 regarding additions to the legend for Fig. 7.

October 21, 2024

RE: GENETICS-2024-307563

Prof. Robert A. Drewell
Clark University
Biology
950 Main Street
Worcester, Massachusetts 1366

Dear Robert:

Congratulations! I am delighted to inform you that your manuscript entitled "Two coacting shadow enhancers regulate twin of eyeless expression during early *Drosophila* development" is acceptable for publication in GENETICS. In review of your submission, I feel that you were responsive to the concerns from all reviewers and so I did not send out for re-review. Many thanks for submitting your research to the journal.

To Proceed to Production:

1. Format your article according to GENETICS style, as discussed at <https://academic.oup.com/genetics/pages/general-instructions>, and upload your final files at <https://genetics.msubmit.net>.
2. Your manuscript will be published as-is (unedited-as submitted, reviewed, and accepted) at the GENETICS website as an Advanced Access article and deposited into PubMed shortly after receipt of source files and the completed license to publish. Please notify sourcefiles@thegsajournals.org if you do not wish to publish your article via Advanced Access.
3. We invite you to submit an original color figure related to your paper for consideration as cover art. Please email your submission to the editorial office or upload it with your final files. You can submit a small-sized image for evaluation, and if selected, the final image must be a TIFF file 2513px wide by 3263px high (8.375 by 10.875 inches; resolution of 600ppi). Please avoid graphs and small type.

If you have any questions or encounter any problems while uploading your accepted manuscript files, please email the editorial office at sourcefiles@thegsajournals.org.

Sincerely,

Pamela Geyer
Associate Editor
GENETICS

Approved by:
Karen Arndt
Senior Editor
GENETICS

note: Please add jnls.author.support@oup.com and genetics.oup@kwglobal.com (or the domains @oup.com and @kwglobal.com) to your email program's "safe senders" list. You will be contacted by both at various points during the production process.

Review comments (if applicable):